# Robustness of the microtubule network self-organization in epithelia

**Aleksandra Z Płochocka[1], Miguel Ramirez Moreno[2], Alexander M Davie[3], Natalia A Bulgakova[2†]\*, Lyubov Chumakova[3†]\***

[1]Center for Computational Biology, Flatiron Institute, New York, United States; [2]Department of Biomedical Science, The University of Sheffield, Sheffield, United Kingdom; [3]Maxwell Institute for Mathematical Sciences, School of Mathematics, Edinburgh University, Edinburgh, United Kingdom

**Abstract** Robustness of biological systems is crucial for their survival, however, for many systems its origin is an open question. Here, we analyze one subcellular level system, the microtubule cytoskeleton. Microtubules self-organize into a network, along which cellular components are delivered to their biologically relevant locations. While the dynamics of individual microtubules is sensitive to the organism's environment and genetics, a similar sensitivity of the overall network would result in pathologies. Our large-scale stochastic simulations show that the self-organization of microtubule networks is robust in a wide parameter range in individual cells. We confirm this robustness *in vivo* on the tissue-scale using genetic manipulations of *Drosophila* epithelial cells. Finally, our minimal mathematical model shows that the origin of robustness is the separation of time-scales in microtubule dynamics rates. Altogether, we demonstrate that the tissue-scale self-organization of a microtubule network depends only on cell geometry and the distribution of the microtubule minus-ends.

**\*For correspondence:**
n.bulgakova@sheffield.ac.uk
(NAB);
lchumakova@gmail.com (LC)

[†]These authors contributed
equally to this work

**Competing interests:** The
authors declare that no
competing interests exist.

**Reviewing editor:** Raymond E
Goldstein, University of
Cambridge, United Kingdom

## Introduction

The correct positioning of intracellular components such as proteins and organelles is critical for correct cellular function (*St Johnston and Ahringer, 2010*; *Ryder and Lerit, 2018*). These components are transported to their biologically relevant locations by motor proteins moving along the cytoskeleton (*Gagnon and Mowry, 2011*; *Kapitein and Hoogenraad, 2011*; *Franker and Hoogenraad, 2013*), or through active diffusion often dependent on these motors (*Drechsler et al., 2017*; *Colin et al., 2020*). Therefore, the direction of cytoskeleton filaments guides the direction and efficiency of intracellular transport. One common type of cytoskeleton used for transport is microtubules (*Franker and Hoogenraad, 2013*). These are highly dynamic unstable polymers that switch between periods of growth and shrinkage. During the growth phase, GTP-tubulin dimers are added to the microtubule plus-end forming a GTP-cap. GTP-tubulin stochastically hydrolyzes into GDP-tubulin, and the loss of the GTP-cap results in a catastrophe: a switch to depolymerization (*Bowne-Anderson et al., 2013*). Microtubule dynamical properties are influenced by a wide range of both internal and environmental factors. For example, the dynamics and number of individual microtubules in a cell depend on the expression of particular plus- and minus-end binding proteins (*Akhmanova and Steinmetz, 2015*), the interaction between microtubules is affected by the presence of crosslinking and motor proteins (*Kapitein and Hoogenraad, 2015*), and the stability of the microtubule network is affected by external factors, e.g. temperature, which changes microtubule rigidity (*Kawaguchi and Yamaguchi, 2010*).

Animal cells take multiple shapes and forms depending on their function, ranging from neuronal cells with meter-long projections, to epithelial cells, to migrating ameboidal leukocytes. Therefore, it is not surprising that microtubule systems similarly acquire a multitude of organizations. In

undifferentiated cells, microtubules form a radial array with minus-ends at a microtubule organizing center at a centrosome. Upon cell differentiation, microtubules are reorganized into non-centrosomal arrays of varying geometry, ranging from unidirectional bundles (e.g. axons), to bidirectional microtubule systems (e.g. subapical microtubules or microtubules in dendrites; *Muroyama and Lechler, 2017*). Microtubule self-organization in unidirectional or radial microtubule systems has been extensively studied (e.g. *Surrey et al., 2001*; *Nédélec et al., 2003*; *Dehmelt, 2014*; *Kapitein and Hoogenraad, 2015*). Directionality and alignment of these networks depend on several interconnected factors. These include the localization of microtubule minus-ends, which could be concentrated at a single location (microtubule organizing center), distributed uniformly at the cell surface, or targeted to specific locations, for example, to the sites of cell-cell contacts (*Muroyama and Lechler, 2017*). The other two factors affecting microtubule network organization are the geometrical constraints of a cell, for example, in a long and thin axon or dendrite, microtubules can only grow in specific directions; and the presence of crosslinking and motor proteins, which promote assembly and orient microtubule bundles (*Zemel and Mogilner, 2008*; *Zemel and Mogilner, 2009*). Bidirectional microtubule networks have an additional level of complexity, as several other factors independently contribute to their organization: the dynamics of individual microtubules, their interactions, and the often dispersed distribution of minus-ends.

In this paper, we explore the self-organization of bidirectional microtubule networks, which are particularly common in differentiated epithelial cells – one of the four fundamental tissue types found in all animals (*Gilbert et al., 1991*; *Bulgakova et al., 2013*; *Muroyama and Lechler, 2017*; *Tateishi et al., 2017*). Here, our focus is the mixed-orientation microtubules just under the apical surface of epithelial cells that are seeded from sites of cell-cell adhesion at the cell periphery (*Toya and Takeichi, 2016*), and not unidirectional apical-basal microtubules. The advantage of this system is that it is physically constrained in space, which allows us to model it as quasi-2d (*Gomez et al., 2016*). We define microtubule self-organization as the degree of alignment of individual microtubule filaments with each other, which we quantify using a length-weighted microtubule angle distribution. Hence, more peaked (or flat) microtubule angle distributions correspond to more aligned (or disordered) microtubule networks. In this work, the microtubule angle distribution describes long-term steady state of the system. In particular, in both the stochastic simulations and analytical model, it was computed as a long time average, and *in vivo* it was computed using averaging cells of the same eccentricity within a tissue. Using other metrics, for example, the 2d nematic order parameter $S_2$, which is a quantitative measure of the orientational ordering (*de Gennes and Prost, 1993*), was prohibitive, since the microtubule density was too low in both our stochastic simulations and experimental images. In contrast, using microtubule angle distribution allowed for accurate comparison between simulations, analytical, and experimental results. Furthermore, it was not possible to perform a systematic analysis of microtubule self-organization dependence on many parameters of microtubule dynamics *in vivo*. This was due to, first, extremely large number of combinations of individual dependencies, which goes beyond the microtubule network itself; and second, altering the microtubule network has profound consequences on processes relying on microtubules, and thus, on cellular functions. We, therefore, analyzed our system via mathematical modeling and validated the model predictions *in vivo*.

Various modeling approaches have been used for describing microtubule self-organization. However, many of them are specific to a particular tissue. In plants, it was shown that microtubule zipping strongly affects their self-organization (*Tindemans et al., 2010*), and that tension can have a non-negligible effect on stabilizing microtubules (*Hamant et al., 2019*). In larger cells, such as *Drosophila* oocytes, microtubule nucleation at the cortex was shown to be important (*Khuc Trong et al., 2015*). Models that include the hydrodynamic effect of the cytoplasm and molecular motors' effect on microtubule self-organization are summarized in *Shelley, 2016* and *Belmonte et al., 2017*. Our published stochastic model successfully recapitulates the organization of microtubule networks in various epithelial cells (*Gomez et al., 2016*). It is a minimal *in silico* 2d-model, in which the microtubules are seeded on the cell periphery, grow stochastically to capture the dynamic instability (as in *Peskin, 1998*), and follow geometric interaction rules.

Here, we use this stochastic model for simulations exploring the typical parameter space of microtubule dynamics, discovering that the average microtubule self-organization is robust. We confirm the robustness *in vivo* using genetic manipulations of epithelial cells in the model organism *Drosophila*. Finally, we build a minimal probabilistic model, which accurately predicts the experimental results

and reveals that the reason for robustness is the separation of time-scales in microtubule dynamics. This model shows that the details of microtubule dynamic instability are irrelevant for microtubule self-organization within their biologically relevant ranges and that the only biological quantity beyond cell shape that affects microtubule alignment is the minus-end distribution. Therefore, we demonstrate the extreme robustness of bidirectional quasi-2d microtubule self-organization, which can be explained by simple mathematical rules. This suggests the general applicability of our findings to quasi-2d microtubule networks, and provides a foundation for future studies.

## Results

### A conceptual geometric model accurately predicts *in vivo* alignment

Cells of the *Drosophila* epidermis elongate during stages 12–15 of embryonic development, changing their eccentricity from 0.7 to 0.98 (*Figure 1A* and *Gomez et al., 2016*). As cells elongate, initially randomly oriented microtubules become gradually aligned (*Gomez et al., 2016*). The simplest thought experiment to visualize how cell elongation translates into microtubule alignment is the following. Imagine a 'hairy' unit-circle on the $(x, y)$ plane, where 'hairs' are microtubules. Turn it inside out (*Figure 1B*). The microtubules are randomly pointing inside the ball, representing the absence of microtubule alignment in non-elongated cells; at each microtubule minus-end on the cell boundary, the mean microtubule direction is normal to the cell boundary. We now deform both the cell and the filament directions by stretching the cell uniformly from its center in the $y$-direction by a factor $b$. This results in an ellipse with eccentricity $e(b) = \sqrt{1 - b^{-2}}$, where the minus-end positions move proportionally to deformation, the filament directions point towards the cell center and their lengths remain unaltered. Mathematically, the distribution $\rho(\theta)$ of microtubule angles $\theta \in [0, 180]$ changes from uniform, $\rho(\theta) = 1/180$, to the angle distribution we call *hairyball* distribution, $\rho_{HB}(\phi)$, which is the inverse Jacobian of the stretching mapping $F : \theta \to \phi$, where $\theta, \phi \in [0, 180]$, given by $\tan(\phi(\theta)) = b \tan(\theta)$,

$$\rho_{HB}(\phi) = \frac{1}{M} \frac{1}{\sin^2 \phi + b^2 \cos^2 \phi}, \quad \phi \in [0, 180], \tag{1}$$

where $M = 180/b$ is the normalization constant. This result gives a surprisingly good agreement with the experiment (*Figure 1C*), especially considering that this model does not take into account the underlying biological processes, for example, microtubule dynamics. Therefore, while a detailed mathematical model is required to understand how various biological processes control microtubule

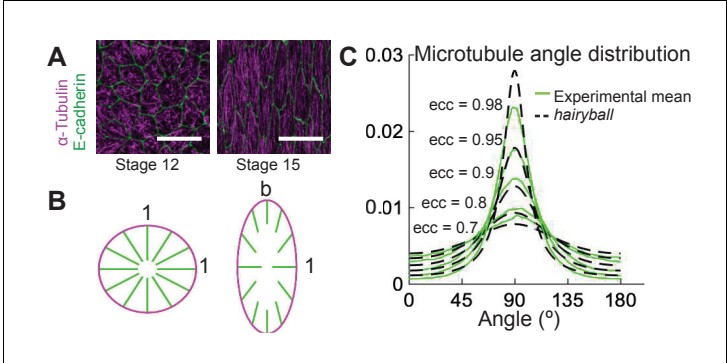

**Figure 1.** A conceptual geometric model of microtubule self-organization. (A) *Drosophila* epidermal cells elongate between stages 12 and 15 of embryonic development, during which the microtubules become more aligned. The scale bar is 10 μm. (B) Stretching a circular cell of radius 1 by a factor $b>1$ deforms the initially uniform microtubule angle distribution into the *hairyball* distribution, *Equation 1*. (C) The experimental microtubule angle distribution (*green*) from stage 12–15 *Drosophila* epidermal cells and the *hairyball* (*dashed black*) are in good agreement up to eccentricity 0.95 . For each eccentricity, the displayed experimental distribution is the mean distribution averaged across cells with the set eccentricity (±0.025 for $ecc = 0.7 - 0.95$ and ±0.005 for $ecc = 0.98$), and is produced as described in the Materials and methods. The number of cells per eccentricity ranged from 348 to 2748.

alignment, the *hairyball* angle distribution formula in *Equation 1* provides a valuable shortcut for the analysis of biological data and parameterizations of microtubule angle distribution, where it can be used to fit data with one parameter – the 'effective aspect ratio' $b$, and, therefore, eccentricity.

## Stochastic simulations demonstrate robustness of microtubule self-organization for a wide range of parameter values

We now explore how microtubule self-organization depends on dynamics and interactions of individual microtubules *in silico*. To this end, we use the same model set-up as previously published (*Gomez et al., 2016*), as this stochastic model recapitulated microtubule self-organization observed *in vivo*. To focus the study on the role of microtubule dynamic instability and interactions and reduce the number of free parameters, we kept the density of the microtubule minus-ends on the cell boundary uniform. We ran the simulations on cells with fixed shapes, since *in vivo* the cell shape evolves on a longer time-scale (hours) as compared to the time required for the microtubule network to stabilize (several minutes) (*Gomez et al., 2016*). We chose the cell shape to be an ellipse since we demonstrate below that the averaged experimental cell shape is an ellipse as well. Finally, the cell eccentricity range in the simulations, 0.7 – 0.98, mimicked the experimental one.

To capture the dynamic instability, we model microtubules as follows (*Figure 2A*). Since the microtubule width (24 nm) is much smaller than the typical cell size (2–10 μm) (*Bulgakova et al., 2013*), we model microtubules as 1d filaments. They are composed of equal length segments, representing microtubule dimers, whose dynamics is governed by a continuous time Markov chain (*Figure 2A*; *Peskin, 1998*; *Gomez et al., 2016*). The microtubule grows (*polymerizes*) at the rate $\alpha$, and shrinks (*depolymerizes*) at the rate $\beta > \alpha$; it switches from the polymerizing to depolymerizing state at the *catastrophe* rate $\beta'$; and undergoes the reverse switch at the *rescue* rate $\alpha'$. These

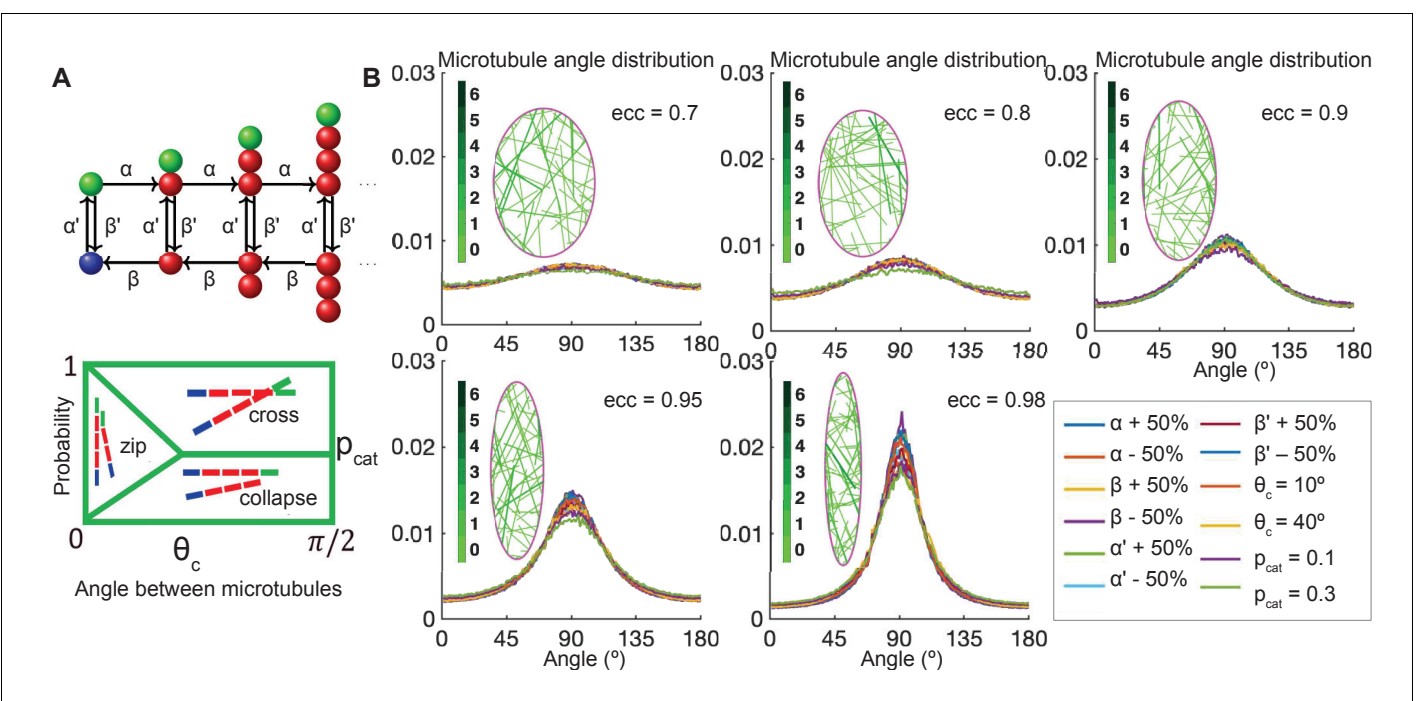

**Figure 2.** Stochastic simulations demonstrate robustness of microtubule self-organization. (A, top) Markov chain: each microtubule grows from the minus-end (*blue*) at the rescue rate $\alpha'$, polymerizes at the rate $\alpha$ if it is stable with a T. GTP cap (*green*), undergoes catastrophe losing the T.GTP cap at the rate $\beta'$, depolymerizes at the rate $\beta$, and regains the T.GTP cap with the rescue rate $\alpha'$. (A, bottom) Parameterization of the effect of crosslinking proteins on microtubule-microtubule interactions, where the probability of an interaction scenario depends on the angle between microtubules. Here $\theta_c$ is the critical angle of zipping, and $p_{cat}$ is the probability of catastrophe. (B) Sensitivity analysis of the microtubule angle distribution. Left: Snapshots of zipping simulations in cells (*magenta*) of different eccentricities; for the base-level parameters $(\alpha, \beta, \alpha', \beta') = (1000, 3500, 4, 1)$, $\theta_c = 30$ and $p_{cat} = 0.01$. Interacting microtubules form bundles, the colorbar indicates the number of microtubules in a bundle. Right: The microtubule angle distributions do not vary significantly in a wide parameter range, suggesting robustness of the microtubule self-organization. The distributions are shown for the variations of $(\alpha, \beta, \alpha', \beta')$ as compared to $(1000, 3500, 4, 1)$, variable critical zipping angle $\theta_c$ and probability of catastrophe $p_{cat}$.

dynamic instability rates (apart from $\beta$) depend on the concentration of free tubulin dimers in cytoplasm (*Walker et al., 1988*), which is reported to vary between 30 and 75% of the total tubulin in cells *in vivo* (*Pipeleers et al., 1977*; *Reaven et al., 1977*; *Zhai and Borisy, 1994*). However, after the microtubule network stabilizes, the total amount of tubulin in microtubules, and, therefore, in the cytoplasm, remains approximately constant, leading to approximately constant dynamic instability rates. Since we investigate the statistics of the microtubule network in steady state (see Materials and methods), we use constant microtubule dynamic instability rates throughout the simulation, and set the same rescue rate for a completely depolymerized microtubule as for any non-zero length microtubule. To account for potentially different microtubule network organizations due to the initial tubulin concentration, we investigate a broad range of parameters. As discussed below, we find that the microtubule angle distribution of stabilized microtubule networks is not sensitive to the parameters of the dynamic instability.

To include microtubule interactions with other microtubules and cell boundaries in the model, we parameterize them using the known parameterization in plants (*Dixit and Cyr, 2004*; *Tindemans et al., 2010*) and *Drosophila* cells (*Figure 2A*; *Gomez et al., 2016*), which relies on two parameters – the critical angle, $\theta_c$, and the probability of catastrophe, $p_{cat}$ – as follows. Upon encountering another microtubule at an angle $\theta$, if $\theta \geq \theta_c$, the growing microtubule either undergoes a catastrophe with probability $p_{cat}$ or crosses the microtubule otherwise. If $\theta \leq \theta_c$, it collapses with probability $p(\theta) = \frac{\theta}{\theta_c} p_{cat}$, crosses with probability $p(\theta) = \frac{\theta}{\theta_c}(1 - p_{cat})$, and bends to change its direction and continues to grow parallel to the existing microtubule otherwise (the microtubule is said to *zip*). Upon reaching a cell boundary at an angle $\theta \leq \theta_c$, the microtubule zips along it with probability $p(\theta) = 1 - \frac{\theta}{\theta_c}$, and depolymerizes otherwise (*Gomez et al., 2016*). These rules for microtubule interactions were originally inspired by the well-established induction of catastrophe when a microtubule grows against a barrier (*Janson et al., 2003*). The role of cell borders as barriers for microtubule growth in epithelial cells is supported by the observations that microtubules buckle at the cell cortex (*Singh et al., 2018*) and that the microtubule catastrophes at the cell boundaries are angle-dependent *in vivo* (*Gomez et al., 2016*). We envision that the exact critical angle, $\theta_c$, and the probability of catastrophe, $p_{cat}$, depend on the presence of specific crosslinking and motor proteins that promote microtubule bundling and stability (*Yan et al., 2013*; *Takács et al., 2017*). Finally, we did not include in the model the effect of microtubules sliding along each other promoted by crosslinking proteins. Sliding has a profound effect on microtubule self-organization in diverse other systems ranging from oocytes to neurons (*Zemel and Mogilner, 2008*; *Lu et al., 2016*; *Winding et al., 2016*). However, we assume this effect to be secondary in our system, since microtubule minus-ends being anchored at the cell boundary (as opposed to being free) prevents sliding. Instead, the leading cause of microtubule self-organization in our system is that microtubule plus-ends are dynamic (as opposed to stabilized). This allows microtubules to 'sense' the scale of the cell: when a microtubule grows toward a cell boundary, upon reaching it, the microtubule either continues growing parallel to it or undergoes a catastrophe.

This setup allows us to investigate a broad range of biologically relevant microtubule dynamics scenarios: in an organism, the dynamic instability parameters are linked to the expression of plus- and minus-end binding proteins and severing factors, while the interaction parameters $\theta_c$ and $p_{cat}$ are linked to the presence of crosslinking and motor proteins, as described above, and temperature-dependent microtubule rigidity (*Kawaguchi and Yamaguchi, 2010*). The relation of the model parameter values to their dimensional equivalents is as follows. The typical observed microtubule growth speed is 0.15 μm/s (*Gomez et al., 2016*). Expressing it as $\alpha \times d \times R$, where $\alpha = 1000$ is the non-dimensional base growth rate, $d = 8.2$ nm is the height of one dimer, and $R$ is the dimensionality coefficient, we find $R$ to be 0.0183 s$^{-1}$. Therefore, the dimensional rates are: the microtubule growth speed $\alpha_{dim,speed} = 0.15$ μm/s; shrinking speed $\beta_{dim,speed} = 0.52$ μm/s; rescue rate $\alpha'_{dim} = 0.07316$ s$^{-1}$; and catastrophe rate $\beta'_{dim} = 0.01829$ s$^{-1}$.

We varied $(\alpha, \beta, \alpha', \beta')$ independently, each from 0.5 to 1.5 times the base-line value $(\alpha, \beta, \alpha', \beta') = (1000, 3500, 4, 1)$. We, therefore, tested the following range of dimensional rates: the microtubule growth speed $\alpha_{dim,speed} = 0.075 - 0.225$ μm/s; shrinking speed $\beta_{dim,speed} = 0.26 - 0.78$ μm/s; rescue rate $\alpha'_{dim} = 0.0366 - 0.1097$ s$^{-1}$; and catastrophe rate $\beta'_{dim} = 0.0091 - 0.0274$ s$^{-1}$. These are in the biologically relevant range, since *in vivo* the parameters of microtubule dynamics depend widely on the cell type, with the reported ranges of growth being 0.05–0.5 μm/s; shrinking 0.13–0.6 μm/

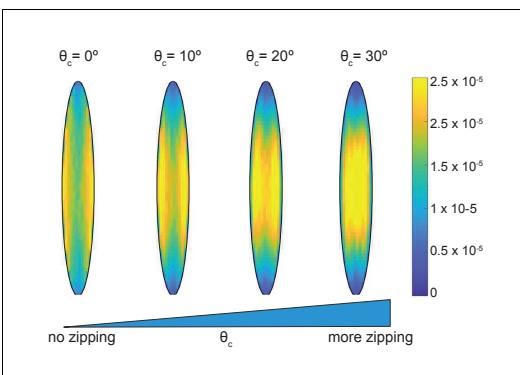

**Figure 3.** Time-averaged microtubule density depends on the strength of zipping. The cell eccentricity was $ecc = 0.98$; the other parameters were kept at their base-level values.

s; rescue 0.01–0.17 s$^{-1}$; and catastrophe 0.003–0.08 s$^{-1}$ (*Rogers et al., 2002*; *Komarova et al., 2005*; *Bulgakova et al., 2013*). We further varied $\theta_c$ between 0° and 40° and $p_{cat}$ between 0.01 and 0.3, where $\theta_c = 10, 40$ and $p_{cat} = 0.1, 0.3$ are reported in *Figure 2B*. As the strength of microtubule interaction increases with $\theta_c$, the case $\theta_c = 0$ corresponds to non-interacting microtubules, which would occur, for example, in the absence of crosslinkers.

The simulations gave an unexpected result (*Figure 2B*) that the microtubule angle distribution varied only slightly, suggesting that in this model microtubule self-organization is robust in individual cells. In particular, it does not depend on the details of dynamic instability and the strength of microtubule interaction, which could range from strong (at $\theta_c = 40$) to non-existent (at $\theta_c = 0$). Despite the robustness of microtubule self-organization measured via the microtubule angle distribution, we found that some features of the microtubule network varied spatially inside cells (*Figure 3*). In the absence of zipping (at $\theta_c = 0$), the microtubule density per unit area was the highest near the longer cell sides, and these peaks shifted toward the cell center with increasing zipping (higher $\theta_c$). Altogether, our model suggests that the microtubule angle distribution for any given cell is robust, while other microtubule network characteristics could vary spatially inside it.

## *In vivo* manipulations of microtubule dynamics and stability alter microtubule density but not alignment

To test *in vivo* the robustness of microtubule self-organization predicted by the stochastic simulations, we examined how changes in microtubule dynamics and stability affect the organization of subapical microtubules in cells of the *Drosophila* embryonic epidermis, where microtubules are constrained to the thin 1 μm apical layer of the cell and grow in the plane of the adhesion belt (*Gomez et al., 2016*). Using genetic manipulations, we could either increase the catastrophe rate $\beta'$, simultaneously increase the catastrophe rate $\beta'$ and shrinkage rate $\beta$, or reduce the number of minus-ends, therefore reducing the density of the network and thus encounters and zipping between microtubules. In particular, we increased $\beta'$ by overexpressing a dominant-negative variant of End Binding protein 1 (EB1-DN), a form which increases the number of catastrophes without changing other parameters of microtubule dynamics (*Bulgakova et al., 2013*; *Komarova et al., 2005*) or increased both $\beta'$ and $\beta$ by overexpressing the protein Spastin, which severs and disassembles microtubules (*Bulgakova et al., 2013*; *Roll-Mecak and Vale, 2005*; *Komarova et al., 2005*). These proteins were overexpressed using the UAS-Gal4 system, in which the Gal4 protein expressed from a tissue-specific promoter induces overexpression of the protein of interest by binding the Upstream Activating Sequence (UAS) (*Brand and Perrimon, 1993*). Specifically, we used *engrailed*::Gal4, which drives expression in stripes along the dorso-ventral axis of embryos, which correspond to posterior halves of each segment (*Figure 4A*). In this instance, we avoided abolishing all microtubules by using mild Spastin overexpression (*Figure 4B*). Overexpression of a CD8-Cherry protein, which does not alter microtubules, was used as a control. We also reduced the number of minus-ends using a null mutation of the minus-end capping protein Patronin (*Nashchekin et al., 2016*), one of the best characterized proteins that protects microtubule minus-ends (*Goodwin and Vale, 2010*). We used zygotic mutants - embryos carrying two mutant alleles of *Patronin* but produced by heterozygous mothers with one wild type allele. Therefore, some Patronin protein is present in homozygous mutant embryos due to maternal contribution - the protein supplied by mothers into eggs, leading to subapical microtubules being reduced but not abolished. Finally, we altered microtubule interactions by manipulating the motor protein Kinesin-1 and the microtubule-actin crosslinker Shortstop (Shot). Kinesin-1 crosslinks microtubules (*Yan et al., 2013*), thus its downregulation simulates

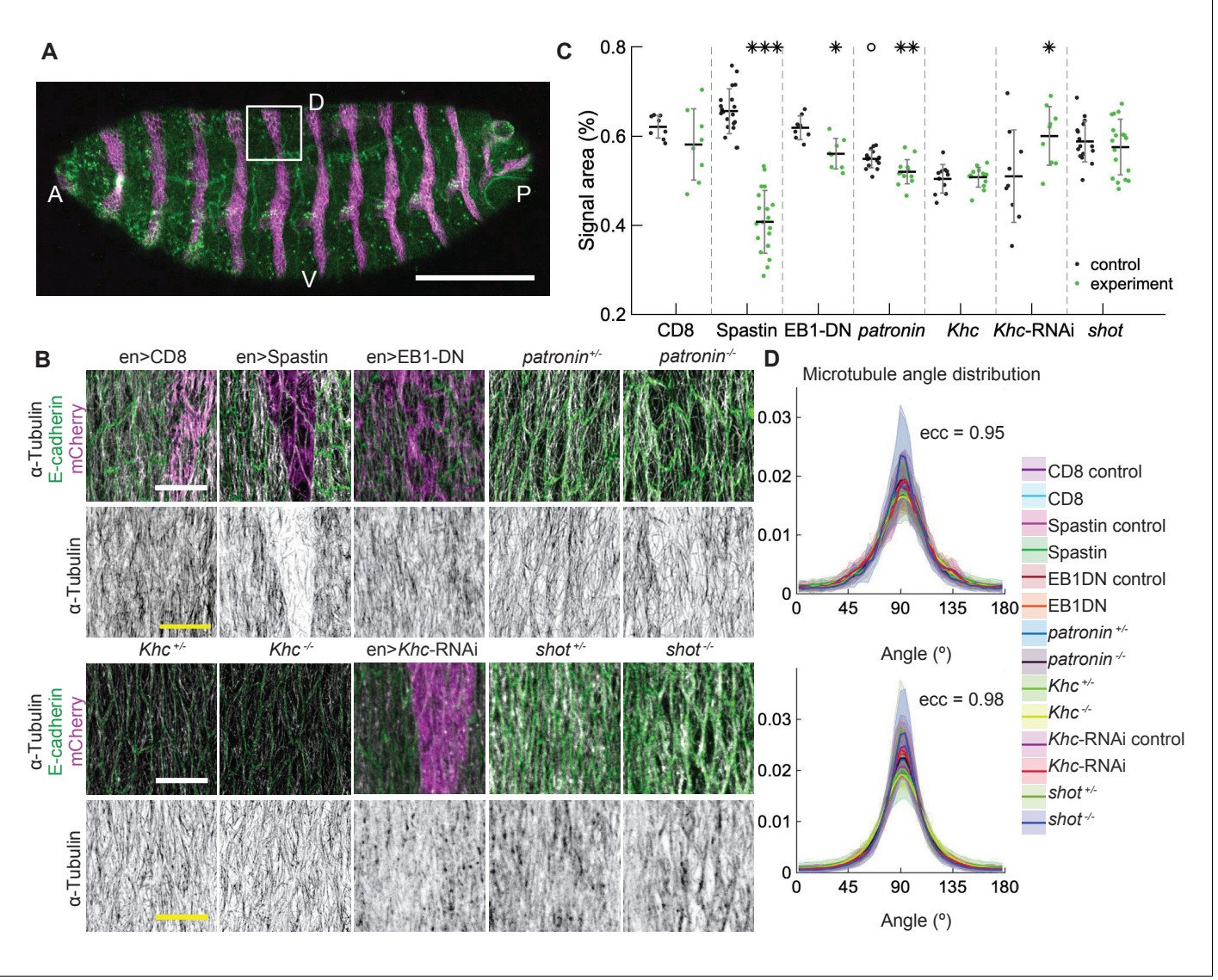

**Figure 4.** Changes to microtubule dynamics and stability do not affect their alignment in the *Drosophila* embryonic epidermis *in vivo*. (**A**) An overview image of a *Drosophila* embryo at stage 15 of embryonic development with cell outlines visualized with E-cadherin (*green*), and *engrailed*-expressing stripes are visualized by direct fluorescence of mCherry (*magenta*). The white square demonstrates the area used for analysis on microtubule organization as shown in (**B**). Anterior (A), posterior (P), dorsal (D), and ventral (V) sides of embryos are labeled. Scale bar - 100 μm. (**B**) Apical view of epidermis from control embryos and with altered microtubules. Top, left-to-right: embryos with CD8-Cherry (control), Spastin (Spas), and EB1-DN expressed using *engrailed*::Gal4, heterozygous *Patronin* ⁺ᐟ⁻, and homozygous *Patronin* ⁻ᐟ⁻ embryos. Bottom, left-to-right: heterozygous *Khc* ⁺ᐟ⁻ and homozygous *Khc* ⁻ᐟ⁻ embryos, embryos expressing *Khc*-RNAi, and heterozygous *shot* ⁺ᐟ⁻ and homozygous *shot* ⁻ᐟ⁻ embryos. Cells expressing CD8-Cherry and EB1-DN are visualized by direct fluorescence of mCherry fused to respective proteins, whereas cells expressing Spastin and *Khc*-RNAi are visualized by coexpression of CD8-Cherry (*magenta*). Cell outlines were visualized by immunostaining against E-cadherin or native fluorescence of E-cadherin-GFP for *Khc*-RNAi (*green*, top row), and microtubules by immunostaining against α-Tubulin (*white*, top row; *black*, bottom row). Scale bar - 10 μm. (**C**) Quantification of microtubule density in each genotype. Internal controls (cells not expressing *engrailed*::Gal4) were used for CD8-Cherry, Spastin, EB1-DN, and *Khc*-RNAi. For *Patronin*, *Khc* and *shot*, heterozygous and homozygous embryos were compared. *** - $p<0.0001$, ** - $p<0.001$, * - $p \leq 0.01$ in comparison to respective control; ° - $p<0.01$ in comparison to CD8-Cherry control. (**D**) The microtubule angle distributions for each eccentricity (±0.025 for $ecc = 0.95$ and ±0.005 for $ecc = 0.98$) do not differ between all genotypes and relative to controls. The distributions – mean (solid line) with standard deviation (shading) – are produced by binning cells in each genotype by eccentricity with the number of binned cells from 32 to 833.

reduced microtubule zipping. In contrast, Shot stabilizes microtubules and crosslinks them with actin cytoskeleton (*Takács et al., 2017*). We used the same *engrailed*::Gal4 to knockdown the Kinesin-1 using the expression of interfering RNA (RNAi) against its heavy chain (*Khc*-RNAi), and complemented this knockdown by using zygotic mutants for *Khc* and *shot*.

We quantified how the above manipulations altered the organization of the microtubule network in cells by obtaining images of microtubules stained with an antibody recognizing α-Tubulin, and analyzing them on a cell-by-cell basis. From these images, we obtained two types of information about microtubule organization (see Materials and methods). First, we determined how the manipulations altered the amount of microtubules in cells by quantifying the percent of the cell area covered by α-Tubulin signal. This measure is a good proxy for the number of microtubules in cells, but it is also sensitive to local arrangements of microtubules – for example, less clustered microtubules due to reduced crosslinking produce a larger area of α-Tubulin signal, as shown below. Therefore, the interpretation of changes in α-Tubulin signal area is done in relation to the known effects of each modification. Second, we determined if the genetic manipulations altered the microtubule alignment. To do this, we determined the direction and magnitude of change of the α-Tubulin signal at each position within the cell (see Materials and methods and *Gomez et al., 2016*), and produced the microtubule direction distributions.

We focused on a late stage of *Drosophila* embryo development (stage 15), as the amounts of protein expressed using UAS-Gal4 system increases, whereas amounts of protein supplied from mothers into zygotic mutants (the maternal contribution) decrease with the progress of embryonic development. The overexpression of both Spastin and EB1-DN reduced the area of α-Tubulin signal in cells (p-values $p<0.0001$ and $p = 0.003$, respectively, *Figure 4B–C*), consistent with their functions. Similarly, the area of α-Tubulin signal was reduced in heterozygous *Patronin*$^{+/-}$ embryos in comparison to wild-type controls (p-value $p = 0.02$, *Figure 4A–B*), and even further reduced in homozygous *Patronin*$^{-/-}$ embryos (p-values $p = 0.0003$ and $p = 0.002$ in comparison to wild-type control and heterozygous siblings, respectively, *Figure 4B–C*). This result is consistent with the dose-dependent protection of the microtubule minus-ends by Patronin. In contrast, zygotic loss of Kinesin-1 did not alter the area of α-Tubulin signal as compared to heterozygous siblings (*Figure 4B–C*). In cells both hetero- and homozygous for the Kinesin-1 mutation, this area was the same and significantly smaller than that in controls presented above. The same signal area was measured in two independent experimental repeats (*Appendix 1—figure 1*), which highlights both the reproducibility of our approach and variability of this measure in response to genetic background, unlike the microtubule angle distribution. In contrast, the *Khc*-RNAi led to a slightly elevated signal area ($p = 0.01$, *Figure 4B–C*), which is in agreement with both the presence of less clustered microtubules and the role of this motor protein in crosslinking. We therefore suggest that the absence of detectable difference in α-Tubulin signal area between cells hetero- and homozygous for the Kinesin-1 mutation is either due to rescue by maternally supplied protein or the use of internal controls allowing for more sensitive detection of differences. Finally, although the average α-Tubulin signal area was not affected in *shot* mutant cells (*Figure 4B–C*), we observed that microtubules seemed more likely to be in close proximity of cell boundaries, similar to the organization reported before (*Takács et al., 2017*). Such organization is predicted to result from reducing the zipping strength in our stochastic simulations (*Figure 3*) and is consistent with the role of the Shot protein in microtubule crosslinking with actin. Despite the observed changes in area and subcellular distribution of α-Tubulin signal, and differences between controls, the microtubule angle distributions did not differ between all genotypes and relative to controls in all cases (*Figure 4D*). Altogether, these results support the robustness of microtubule self-organization despite variable microtubule dynamics and amounts.

To capture a wide range of eccentricities, we used the embryos at different stages of development during which the epidermal cells progressively elongate from eccentricities around 0.7 to 0.98 (stages 12 through 15). To this end, we focused on the genotypes with the strongest effects at stage 15 (Spastin, EB1-DN, and CD8-Cherry, as a control) and used *paired*::Gal4, which although leading to milder overexpression than *engrailed*::Gal4, is expressed in broader stripes along the dorso-ventral axis of embryos. Overexpression of Spastin reduced α-Tubulin signal area in comparison to both neighboring cells, which did not express *paired*::Gal4, and control cells expressing CD8-Cherry (both p-values $p<0.0001$, *Appendix 1—figure 2A-B*). Similarly, the α-Tubulin signal area was reduced in embryos homozygous for *Patronin*$^{-/-}$ in comparison to heterozygous siblings (p-value, $p = 0.04$, *Appendix 1—figure 2A-B*). We suggest that the lack of a difference between heterozygous

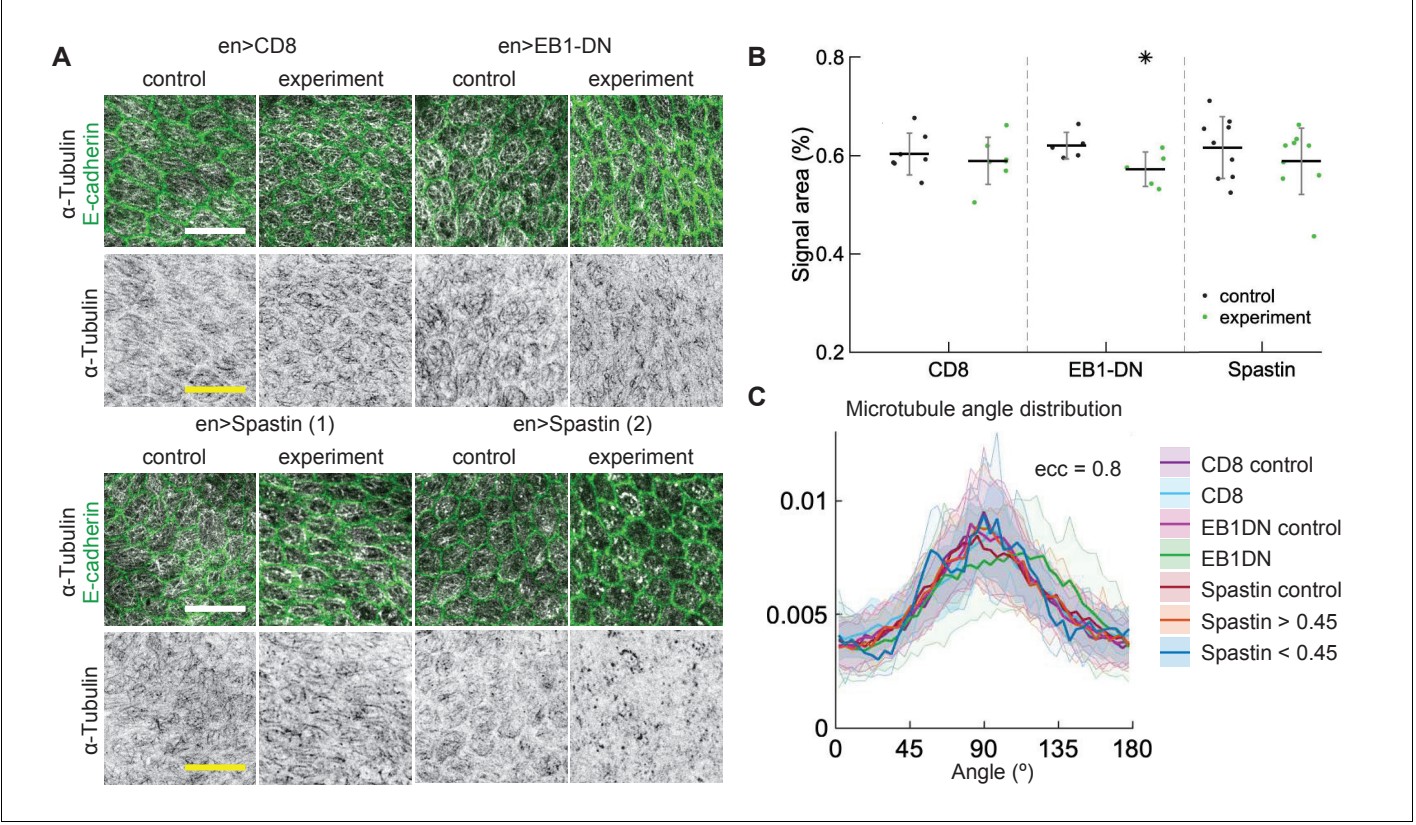

**Figure 5.** Changes to microtubule dynamics and stability do not affect their alignment in *Drosophila* pupal wings *in vivo*. (**A**) Apical view of epithelial cells from pupal wings in control anterior and experimental posterior compartments. Top: pupal wings expressing CD8-Cherry (control) and EB1-DN; bottom: examples of wings expressing Spastin, which leads to a variable phenotype ranging from severed (left) to appearing non-severed (right) microtubules. In all cases, *engrailed*::Gal4 was used. Cells expressing CD8-Cherry and EB1-DN are visualized by direct fluorescence of mCherry directly fused to respective proteins, whereas cells expressing Spastin are visualized by coexpression of CD8-Cherry (not shown). Cell outlines were visualized by native fluorescence of E-cad-GFP (*green*, top rows), and microtubules by immunostaining against α-Tubulin (*white*, top rows; *black*, bottom rows). Scale bar - 10 μm. (**B**) Quantification of microtubule density in each genotype. Internal controls (cells not expressing *engrailed*::Gal4) were used for comparison. * - $p = 0.07$ . (**C**) The microtubule angle distributions for $ecc = 0.80$ do not differ between all genotypes and relative to controls. The distributions – mean (solid line) with standard deviation (shading) – are produced by binning cells in each genotype by eccentricity. Distributions in cells with α-Tubulin signal areas less and greater than 0.45 are shown separately for Spastin. The number of binned cells ranged from 7 to 48.

*Patronin* $^{+/-}$ and the wild-type control observed here might be due to measurements being taken over a broader developmental time, which makes it more difficult to detect changes. Overexpression of EB1-DN did not change the area covered by the α-Tubulin signal per cell (*p*-value, $p = 0.98$, *Appendix 1—figure 2A-B*). This can be explained by either weaker expression of *paired*::Gal4 in comparison to *engrailed*::Gal4, or lower amounts of Gal4 at earlier developmental stages. The microtubule angle distributions for each eccentricity (binned at a particular eccentricity ±0.025) did not significantly differ between all genotypes and relative to controls (*Appendix 1—figure 2C*). Although only two of the above manipulations affected microtubule density, these results further support that microtubule self-organization is indeed robust *in vivo*.

Having found robustness in the embryonic epidermis, we sought to test that the same is true in another epithelial tissue – pupal wing. These cells are elongated to a lesser degree than average epidermis cells in the apical plane, but display similar subapical microtubule networks that align along cells' major axes (*Gomez et al., 2016*). We examined the effects of overexpressing EB1-DN and Spastin in pupal wings, as these had the largest effects on α-Tubulin signal area in embryos, and used CD8-Cherry as a control. We have compared cells in *engrailed*::Gal4-expressing posterior compartments of pupal wings with anterior compartments, which do not express *engrailed*::Gal4 but are otherwise genetically identical. Expression of EB1-DN reduced the area of α-Tubulin signal, although to a lesser extent than in embryos ($p = 0.07$, *Figure 5A–B*). This reduction was not reflected in any

change in the microtubule angle distribution (*Figure 5C*). The expression of Spastin in pupal wings did not produce a consistent phenotype – some wings looked as in control, whereas in others microtubules were clearly severed (*Figure 5A*), which lead to no apparent change in the bulk α-Tubulin signal area ($p = 0.36$, *Figure 5B*). To determine if the microtubule angle distribution was altered in cells with severed microtubules in comparison to non-severed ones, we compared cells with signal areas of α-Tubulin lower than 0.45 with cells where it was greater than 0.45. We found no differences between these two groups of cells (*Figure 5C*). The distributions did not differ from those in cells from anterior compartments, which did not express Spastin, and from cells expressing CD8-Cherry as a control. Altogether, our findings demonstrate that microtubule angle distribution is robust to perturbations in dynamics of individual microtubules, microtubule interaction, and the number of microtubules, in two independent epithelial tissues *in vivo*. As the microtubule angle distributions were produced by averaging cells of the same eccentricity within a tissue, we term this *robustness on the tissue scale*.

## An analytical model shows that microtubule self-organization depends on the cell geometry and minus-end distribution

Given that the microtubule angle distribution both *in silico* and *in vivo* only weakly depends on microtubule interactions, we propose a minimal mathematical model with non-interacting microtubules, which is analytically tractable. Here, independent microtubules cross upon reaching one another and fully depolymerize upon reaching a cell boundary. Their averaged behavior is the average of 1d behaviors of individual microtubules growing from different positions on the cell boundary. While we expect the microtubules to have an overall alignment along the major cell axis due to the high build-in catastrophe rate at cell boundaries, our goal is to obtain the full microtubule angle distribution.

Our model setup is as follows (*Figure 6A*). Consider a convex 2d cell with the boundary parameterized by the arclength-coordinate $\zeta$ increasing in a counter-clockwise direction. From microtubule minus-ends distributed with density $\rho(\zeta)$ on the cell boundary, the microtubules grow into the cell interior at an angle $\theta$ to the cell boundary and at an angle $\varphi$ with respect to the horizontal. Note that the cell shape is fully determined by the function $a(\zeta, \theta)$ – the length of a cross-section that starts at $\zeta$ at an angle $\theta$ with respect to the boundary. When $a(\zeta, \theta)$ is considered as a function of $(\zeta, \varphi)$, we denote it by $\tilde{a}(\zeta, \varphi)$ to avoid confusion. Microtubules undergo dynamic instability by switching between the states of growth, shrinking, catastrophe, and rescue at the rates $\alpha$, $\beta$, $\beta'$, and $\alpha'$, respectively. After fully depolymerizing, they regenerate at the rate $\alpha'$ from the same minus-end but in a new direction at an angle $\theta$ taken from a uniform distribution on $[0, 180]$. Thus, over a large fixed

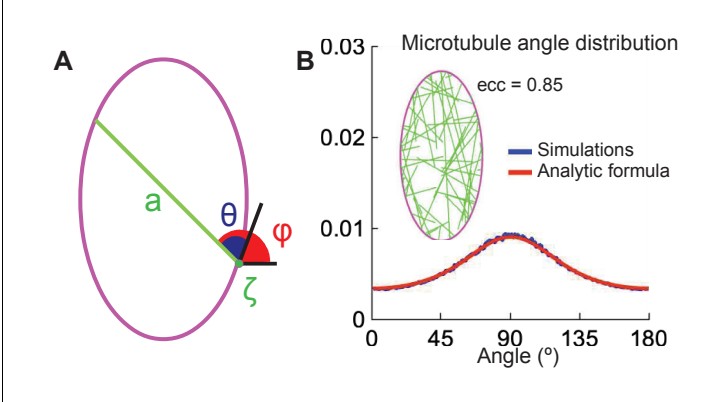

**Figure 6.** The analytical model of microtubule self-organization. (A) Analytical model setup: cell shape is parameterized by the arclength, $\zeta$, along the cell boundary (*magenta*). At the minus-end, $\zeta$, a straight microtubule (*green*) grows at an angle $\theta$ (or $\varphi$) with respect to the cell boundary (or horizontal); its maximum length is the cross-section, $a$, of the cell. (B) Left: snapshot of the simulations with non-interacting microtubules for cells of eccentricity 0.85. Right: the agreement between the microtubule angle distributions given by *Equation 9* (*red*) and the stochastic simulations (*blue*).

time interval $t \in [0, T]$, a microtubule undergoes a large number $N$ of growth and shrinkage lifetimes, which are separated by periods of average duration $1/\alpha'$ when the microtubule has zero length.

The first quantity of interest is the microtubule mean survival time. Since both *in vivo* and *in silico*, the microtubule angle distribution is length-weighted, we include the general case of weighting the mean survival time by a function $\gamma(x)$ of microtubule length $x$. Then the mean survival times, $f(x)$ and $g(x)$, of polymerizing and depolymerizing microtubules of length $x$ satisfy

$$f(x) = (1 - \beta' dt) f(x + \alpha \, dt) + \beta' dt \, g(x) + \gamma(x) \, dt, \tag{2}$$

$$g(x) = (1 - \alpha' dt) g(x - \beta \, dt) + \alpha' dt \, f(x) + \gamma(x) \, dt, \tag{3}$$

where the terms on the right-hand side are the contributions from growing (and shrinking in the $g$ case), switching, and the time increment weighted by $\gamma(x)$, which we specify below. Here, $dt$ is a small time-increment. Expanding *Equation 2-3* in Taylor series and neglecting terms of the second and higher order in $dt$, we obtain that $f(x)$ and $g(x)$ are governed by

$$df(x)/dx = \beta'/\alpha \, (f(x) - g(x)) - \gamma(x)/\alpha, \tag{4}$$

$$dg(x)/dx = \alpha'/\beta \, (f(x) - g(x)) + \gamma(x)/\beta. \tag{5}$$

$$h(x) = f(x) - g(x)$$

$$dh(x)/dx = p h(x) - q \gamma(x), \tag{6}$$

where $p = \frac{\beta'}{\alpha} - \frac{\alpha'}{\beta}$ and $q = \frac{1}{\alpha} + \frac{1}{\beta}$. We assume that $g(a) = 0$, that is, once the microtubule reaches the cell boundary at the length $a(\zeta, \theta)$, it quickly depolymerizes. Finally, for a zero-length microtubule $g(0) = 0$, and hence $h(0) = f(0)$. Note that the only quantity of interest is $f(0) = h(0)$, since it is the lifetime of a microtubule when it starts in a growing state with zero length.

The two choices $\gamma(x) = 1$ and $\gamma(x) = x$ give the solutions for the not-weighted and the length-weighted mean survival times, denoted $f_{nw}(x)$ and $f_w(x)$, respectively

$$f_{nw}(0) = \frac{q}{p}(1 - e^{-pa}), \quad f_w(0) = \frac{q}{p^2}(1 - e^{-pa}(1 + pa)). \tag{7}$$

For each microtubule minus-end location, $\zeta$, the average time between any two re-growths of a microtubule is the sum of the averaged waiting time $1/\alpha'$ and the average of the non-weighted lifetime over all the growth angles $\theta$

$$\frac{T}{N} \approx T_{ave} = \frac{1}{\alpha'} + \int_0^\pi f_{nw}(0) d\theta. \tag{8}$$

Then the average number of lifetimes with direction $(\varphi, \varphi + \delta\varphi)$ with respect to the horizontal is $N\delta\varphi/\pi$, and their contributions to the length-integral is $f_w(0)N\delta\varphi/\pi = (f_w(0)/T_{ave})T/\pi\delta\varphi$. Integrating it over the cell boundary weighted by the density of minus-ends $\rho_m(\zeta)$ and using *Equation 8*, we obtain the length-weighted microtubule angle distribution

$$\rho(\varphi) = \frac{1}{M} \int \frac{1 - e^{-\tilde{a}(\zeta,\varphi)p}(1 + \tilde{a}(\zeta,\varphi)p)}{1 + \frac{q\alpha'}{\pi}\int_0^\pi \frac{1 - e^{-a(\zeta,\theta)p}}{p}d\theta} \rho_m(\zeta)d\zeta, \tag{9}$$

where $M$ is the normalization constant. The cell cross-section is denoted by either $a(\zeta, \theta)$ or $\tilde{a}(\zeta, \varphi)$ depending on its arguments. This analytical prediction matches the stochastic simulations (*Figure 6B*).

The dependency of the resulting microtubule angle distribution on the biological rates reduces to two parameters

$$p = \frac{\beta'}{\alpha} - \frac{\alpha'}{\beta}, \quad \frac{q\alpha'}{\pi} = \frac{\alpha'}{\pi}\left(\frac{1}{\alpha} + \frac{1}{\beta}\right). \tag{10}$$

If both of them are small, $|p| \ll 1$ and $\frac{\alpha'}{\pi}\left(\frac{1}{\alpha}+\frac{1}{\beta}\right) \ll 1$, as is always observed in biological systems (see above), the microtubule angle distribution formula can be significantly simplified. In particular, for small $p$, the distribution in *Equation 9* reduces to

$$\rho(\varphi) = \frac{1}{M} \int \frac{\tilde{a}^2(\zeta,\varphi)}{1 + \frac{q\alpha'}{\pi} \int_0^\pi a(\zeta,\theta)d\theta} \rho_m(\zeta)d\zeta \qquad (11)$$

to leading order, and to

$$\rho(\varphi) = \frac{1}{M} \int \tilde{a}^2(\zeta,\varphi)\rho_m(\zeta)d\zeta, \qquad (12)$$

when, in addition, $\frac{q\alpha'}{\pi} \ll 1/\int_0^\pi a(\zeta,\theta)d\zeta$. This becomes exact in the limit of deterministic microtubules ($\beta = \infty$, $\beta' = 0$). Note that while $p$ is required to be non-negative in models of microtubules on an infinite line (*Peskin, 1998*), our setup does not have this restriction, as the microtubule lifetimes $f_w(0)$ and $f_{nw}(0)$ are positive even for negative $p$. Furthermore, the analytical microtubule angle distribution, *Equation 9*, is independent of the multiplicative change in the minus-end density, $\rho_m(\zeta)$, which would be absorbed into the normalization constant $M$. Only non-trivial changes to the density of minus-ends that vary along the cell boundary affect the microtubule angle distribution.

It is required that $\alpha' \ll \alpha$ and $\alpha' \ll \beta$ for the second parameter in *Equation 10* to be small, and $\beta' \ll \alpha$ for the first parameter to be small as well. This separation of time-scales in microtubule dynamics is observed *in vivo*, as described above, where the rates of polymerization and depolymerization are much higher than those of catastrophe and rescue. Therefore, the microtubule angle distribution depends only on the cell geometry and the minus-end distribution, and the underlying reason for that is the separation of time-scales in microtubule dynamics.

## The analytical model accurately predicts microtubule self-organization given both the experimental cell shape and distribution of microtubule minus-ends

To validate our analytical model's predictions, we sought to compute the analytical microtubule angle distributions in *Equation 9* using experimental cell shape and minus-end density. Note that *Equation 9* predicts that the microtubule angle distributions is robust in each individual cell, where each distribution is determined by the cell geometry and the distribution of minus ends. For example, a cell corner at multicellular junctions (where three or more cells contact) leads to 'corners' in microtubule angle distributions. However, cell shapes in tissue are highly variable, even when they have the same eccentricities. We, therefore, sought to perform comparison between the model and experiment using the averaged values of the microtubule minus end locations and cell shape.

First, we determined the localization of microtubule minus-ends. Since Patronin localizes at the microtubule minus-ends, we analyzed its distribution in epithelial cells in the *Drosophila* embryonic epidermis using Patronin-YFP (*Nashchekin et al., 2016*). As expected, Patronin-YFP is mostly localized at the cell boundaries with few speckles inside cells (*Figure 7A*). We quantified the distribution of Patronin-YFP at the cell boundaries by measuring its asymmetry, namely the ratio of Patronin-YFP average intensity at dorso-ventral borders to that of anterior-posterior borders (see Materials and methods, *Figure 7B*, and *Bulgakova and Brown, 2016*). The asymmetry of Patronin distribution was a linear function of the cell eccentricity (*Figure 7C*), suggesting that Patronin becomes enriched at the dorso-ventral boundaries as the embryo develops and cells elongate. Additionally, when comparing boundaries in cells with similar eccentricities (Stage 15 embryos only), the intensity of Patronin-YFP was decreasing with the border angle relative to the anterior-posterior axis of the embryo (*Figure 7D*). Several lines of evidence support that the observed enrichment of Patronin-YFP at the dorso-ventral boundaries is due to the asymmetry of cell-cell adhesion in these cells. Indeed, microtubule minus-ends were shown to be tethered by cell-cell adhesion in some epithelial cells (*Meng et al., 2008*). Concurrently, E-cadherin, the key component of cell-cell adhesion in *Drosophila* embryonic epidermis, is asymmetrically distributed in stage 15 embryos (*Bulgakova et al., 2013*), with enrichment at the dorsal-ventral borders similar to that of Patronin. Finally, asymmetries of both Patronin and E-cadherin progressively increase from stage 12 to stage 15 of *Drosophila* embryonic development (*Figure 7C and N.A.B.* personal communication).

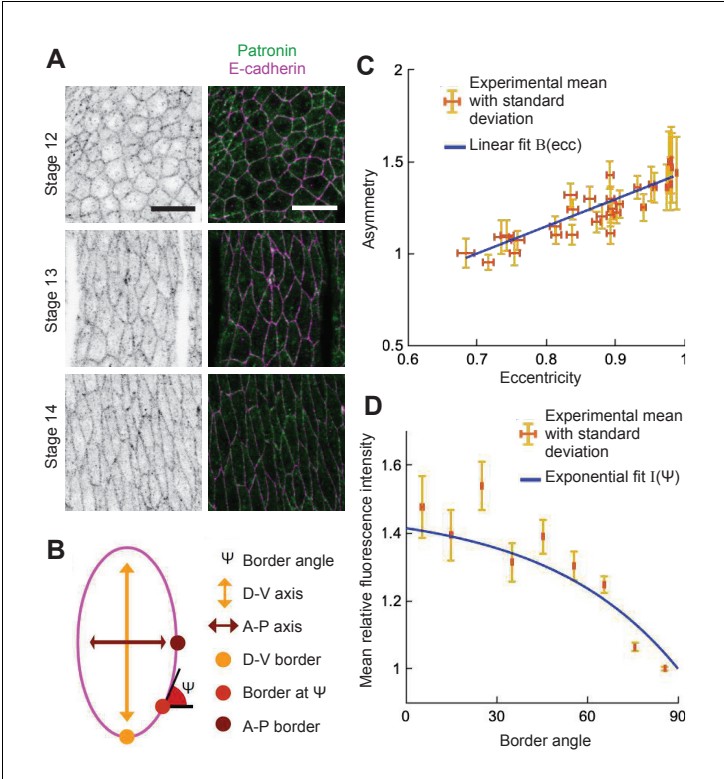

**Figure 7.** Localization of Patronin-YFP in the *Drosophila* embryonic epidermis. (**A**) Apical view of embryonic epidermis at stage 12 (top), 13 (middle), and 15 (bottom), visualized with Patronin-YFP (*gray*, left; *green*, right), and E-cadherin immuno-staining (*magenta*, right). Scale bar - 10 µm. (**B**) Schematic of a cell. (**C**) Asymmetry of Patronin-YFP localization increases linearly with eccentricity. Each dot represents the average values of Patronin-YFP asymmetry and cell eccentricity in a single embryo. Error bars are SD. Solid line visualizes the linear fit of the form $B(ecc) = 1 + C_1(ecc - 0.7)/(0.98 - 0.7)$, $C_1 = 0.4144$. (**D**) Mean relative amounts of Patronin-YFP as a function of the border angle $\psi$ in stage 15 (eccentricity 0.98) *Drosophila* embryonic epidermis, normalized by its value at the vertical long sides (0-10°). Borders were binned at 10° intervals relative to the embryonic anterior-posterior axis, and intensity was averaged for each bin (mean ± SD). The solid line represents the exponential fit for the intensity as $I(\psi) = 1 + (B(0.98) - 1)(1 - e^{-C_2(90-\psi)})/(1 - e^{-90C_2})$, $C_2 = 0.0231$.

To use this in our analytical model, we used least-squares to simultaneously fit the asymmetry data with a linear function of eccentricity, and the normalized intensity of Patronin-YFP with an exponential function of the cell border angle. We imposed a constraint that the asymmetry value at eccentricity 0.98 (Stage 15) is the same as the normalized intensity of Patronin-YFP at the dorso-ventral border (border angle 0°). The resulting formula used in the analytical model is

$$A(ecc, \psi) = 1 + C_1 \frac{ecc - 0.7}{0.98 - 0.7} \left( \frac{1 - e^{C_2(\psi-90)}}{1 - e^{-90C_2}} \right),$$

(13)

where $\psi$ is the cell border angle with respect to the horizontal, $C_1 = 0.4144$ and $C_2 = 0.0231$.

Next, we determined the average cell shape. In tissue, each cell has a unique shape, and cells with the same eccentricities may differ significantly in their geometry. Therefore, to test and validate the analytical solution of microtubule self-organization we have generated masks of epithelial cells in the *Drosophila* embryonic epidermis, which provided us with coordinates of cell boundaries (see Materials and methods). Dividing all cell shapes into groups by eccentricity, we computed the average cell shape for each group as follows. First, the cells were re-centered to have their centers of mass at the origin. We then rotated them so that they are elongated along the vertical axis (the direction of elongation is the first singular vector, see Materials and methods). Finally, we rescaled all the cells to have unit area. The average of the distance from the center of mass in a particular

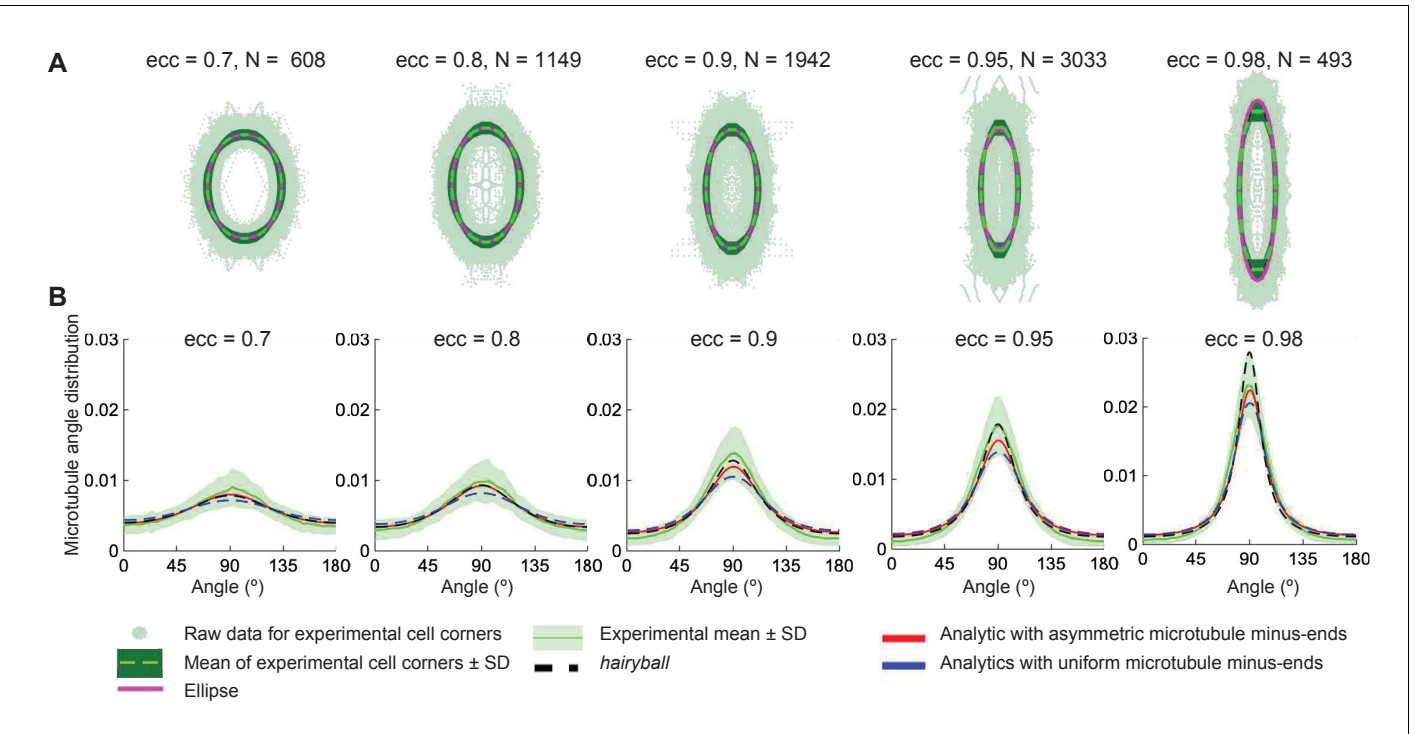

**Figure 8.** Agreement between the analytical and experimental microtubule self-organization on averaged cell-shape. (A) Experimental cell shapes for eccentricities 0.7—0.98. The experimental cell boundary data points (*light green points*), its standard deviation (*darker green envelope*) around the radial mean (*dark green line*). The ellipse (*dashed magenta*) closely approximates the experimental mean shape. The graphs show the corresponding microtubule angle distributions. (B) The analytical distribution with asymmetric minus-ends (*red*) has better agreement with the experimental mean (*dark green line*) (± SD is the *light-green envelope*), comparing to the uniform minus-end density (*blue*); the *hairyball* has good agreement up to eccentricity 0.95 with the experimental mean.

direction to the cell boundary traced the boundary of the averaged cell. Surprisingly, we found that the average cell shape for a given cell eccentricity is an ellipse (*Figure 8A*).

The analytical microtubule angle distribution, *Equation 9*, computed on an averaged cell shape using the experimental minus-end density, *Equation 13*, gives surprising agreement with experimental *in vivo* data (*Figure 8B*). This agreement is better for the case of asymmetric minus-end distribution, comparing to the uniform one, which supports our prediction that the minus-end distribution does indeed influence the microtubule self-organization.

## Discussion

Here, we present several novel findings that describe the fundamental rules underlying self-organization of microtubule networks in epithelial cells. Firstly, we have shown robustness of microtubule self-organization both *in silico* and *in vivo*; secondly, in addition to the known importance of cell shape for microtubule organization (*Gomez et al., 2016*), our minimal analytical model predicted the importance of the asymmetric minus-end distribution as the only other impactful parameter, which we then confirmed *in vivo* in *Drosophila* epidermal cells; and finally, we have shown that this robustness originates from the intrinsic separation of time-scales of microtubule dynamic instability.

### Robustness on the tissue scale

Biologically, the discovered robustness of microtubule organization makes perfect sense, given the fundamental importance of microtubule functions in a cell. Most intracellular trafficking events require microtubules for the delivery of various cellular components to their relevant biological locations by motor proteins (*Hamm-Alvarez, 1998*; *Apodaca, 2001*). This process must be reliable, as mislocalization of cellular components leads to cell death or disease (*Levy et al., 2006*; *Lopes et al.,*

2010; *Baek et al., 2018*). However, this delivery mechanism is highly stochastic, given the microtubule dynamic instability and the dynamics of molecular motors (*Kolomeisky and Fisher, 2007*; *Brouhard, 2015*; *Goodson and Jonasson, 2018*). We suggest that it is the average microtubule angle distribution, which is likely to guide the net outcome of intracellular trafficking and focus on this distribution in this work.

We have shown that going from the subcellular to the tissue scale, the microtubule organization becomes more and more robust. On the subcellular level, some features of microtubule networks are variable, for example, the time-averaged microtubule density per unit area in our stochastic simulations depends on the strength of microtubule interactions. In particular, in systems with strong microtubule interactions, the microtubule density plateaued at the cell center, while in the absence of microtubule interactions, the microtubule density was higher near the cell boundaries, which we also observed experimentally *in vivo* in *shot* mutant cells. Moving to the scale of individual cells, our stochastic simulations and the analytical model showed that the averaged microtubule angle distribution is already robust. This is because the average microtubule self-organization depends to leading order only on the slowly evolving parameters, such as the cell shape and the density of the minus-ends on the cell boundary. However, microtubule self-organization is distinct in individual cells, where individual cell-shape features such as cell-corners at the multicellular junctions are reflected in the microtubule angle distribution, since the latter is proportional to the integral over the cell boundary of the squared cross-section of the cell. Finally, on the multicellular level, the microtubule self-organization is robust on the tissue-scale, since the effect of individual cell shape variability averages out. In particular, while most of the cells of a given eccentricity in biological tissues are polygons, we found that the average cell shape of a particular eccentricity is an ellipse.

One of the findings of our analytical model is that the robustness of microtubule self-organization exists only as long as the microtubule dynamics exhibits a separation of time-scales, $\alpha', \beta' \ll \alpha, \beta$, a rule which is observed in all published data about microtubule dynamic parameters (*Shelden and Wadsworth, 1993*; *Rogers et al., 2002*; *Komarova et al., 2002*; *Komarova et al., 2009*; *Dhonukshe and Gadella, 2003*; *Zilberman et al., 2009*; *Zhang et al., 2011*; *Bulgakova et al., 2013*; *Shaebani et al., 2016*). Our mathematical model shows that if this rule is not observed, the microtubule organization becomes sensitive to changes in these rates. As these rates depend on multiple internal and external factors, such as changes in gene expression and temperature, the microtubule organization would be unpredictable in a cell in a biological tissue. Therefore, breaking this rule will impair cellular function over time, which suggests that any mutations that led to such change were likely to cause cell lethality and did not fix in evolution.

## *Hairyball* distribution

We demonstrated that the microtubule angle distribution is accurately predicted by the *hairyball* distribution. The reason for this excellent agreement with the experimental data remains an open question, as we were unable to show that *hairyball*, which is a result of a conceptual 0-th order model, has any relation to our analytical distribution, neither as an approximation nor as a limiting case. As presented, the *hairyball* distribution does not include the effect of non-uniform microtubule minus-end distribution. We found that including it (as a multiplicative factor in *Equation 1*) does not significantly change the agreement with the experimental data. We suggest that the best use of the *hairyball* distribution is as a simple ad-hoc formula to parameterize the microtubule angle distribution in cells up to eccentricity 0.95 using the single 'effective aspect ratio' parameter. This could prove useful in investigating, for example, correlations and interdependencies between the microtubule network organization (e.g. their overall direction and spread) and dynamic intracellular processes (e.g. signaling and transport).

## The importance of microtubule interactions

While we show that microtubule self-organization does not depend on microtubule interaction, we admit that this is true for the measure of self-organization being the time-averaged microtubule angle distribution. Here, such effects of microtubule interactions as zipping and bundling disappear due to time-averaging of the dynamics. However, from the biological point of view, what matters for an organism is the long-term behavior, because most of the processes such as microtubule-based transport occur on much longer time-scales than microtubule network rearrangements

(*Jankovics and Brunner, 2006*; *Bulgakova et al., 2013*; *Iyer et al., 2019*). Therefore, we hypothesize that it is the microtubule angle distribution that affects tissue behavior on long time-scales. We further hypothesize that such effects as bundling and zipping will affect short-term intracellular transport. For example, the presence of Spastin, the microtubule severing protein, leads to a change in the delivery of the E-cadherin, the protein responsible for the cell-cell adhesion delivered along the microtubule network (*Bulgakova et al., 2013*). A more detailed modeling approach that includes the effect of microtubule-microtubule interaction on intracellular transport is outside the scope of this article and will be considered in a separate publication.

Our *in vivo* experiments show that microtubule-microtubule and microtubule-actin crosslinkers such as Kinesin-1 and Shot, which are known to alter microtubule dynamics and local organization (*Figure 2*; *Jolly et al., 2010*; *Takács et al., 2017*; *Drechsler et al., 2020*), do not alter robustness of average microtubule self-organization. We have recently discovered that molecules localized inhomogeneously in the plane of microtubules (e.g. actin stress fibers) can alter microtubule angle distribution by reorienting the microtubule angle distribution away from the cell's major axis (*Delon and Brown, 2009*; *Płochocka et al., 2019*). However, the average microtubule self-organization in homogeneous environments remains robust in all tested cases.

## Our system is a particular but generalizable scenario

We suggest that our findings are applicable beyond apical microtubules in the *Drosophila* embryonic epidermis, the dynamics of which is quasi-2d. Previously, we demonstrated that a similar relationship between microtubule organization and cell shape is observed in other *Drosophila* epithelia, including cells in pupal wings and ovaries (*Gomez et al., 2016*). Here, we conducted *in vivo* experiments in multiple different genotypes in two tissues: embryonic epidermis and pupal wings. Since the tissue and genetics were varied in these scenarios, the microtubule growth and interaction parameters were altered as well. While this led to noticeable differences in such network properties as microtubule density in different controls and local microtubule organization, the average microtubule self-organization reflected by the microtubule angle distribution remained robust on the tissue scale. There are multiple other instances in which maturing and differentiating epithelial cells develop an apical microtubule meshwork, including cells of mammalian airways and even cells in culture (*Gilbert et al., 1991*; *Herawati et al., 2016*; *Takeda et al., 2018*). Our findings are likely to hold true in these systems as long as in them the microtubules are anchored, the plus-ends are dynamic, and the separation of time-scales holds. Furthermore, these rules are likely to apply to squamous cells, which, despite having specialized apical microtubules, have such a small cell depth that microtubules are constrained within a thin plane similar to that in our experimental model (*Gomez et al., 2012*; *Pope and Harris, 2008*). The validation of our findings in other cell types and other evolutionary divergent organisms, as well as how the discovered robustness of microtubule self-organization ensures the reliability of intracellular transport, are important questions for future research.

## Materials and methods

### Key resources table

| Reagent type (species) or resource | Designation | Source or reference | Identifiers | Additional information |
|---|---|---|---|---|
| Genetic reagent (*Drosophila melanogaster*) | *paired*::Gal4 | Bloomington *Drosophila* Stock Center | RRID:BDSC_1947 | |
| Genetic reagent (*Drosophila melanogaster*) | *engrailed*::Gal4 | Bloomington *Drosophila* Stock Center | RRID:BDSC_30564 | |
| Genetic reagent (*Drosophila melanogaster*) | *UAS*::CD8-Cherry | Bloomington *Drosophila* Stock Center | RRID:BDSC_27392 | |
| Genetic reagent (*Drosophila melanogaster*) | *UAS*::Khc-RNAi | Bloomington *Drosophila* Stock Center | RRID:BDSC_35770 | |
| Genetic reagent (*Drosophila melanogaster*) | *Khc*[8] | Bloomington *Drosophila* Stock Center | RRID:BDSC_1607 | |

*Continued on next page*

*Continued*

| Reagent type (species) or resource | Designation | Source or reference | Identifiers | Additional information |
|---|---|---|---|---|
| Genetic reagent (*Drosophila melanogaster*) | *shot*[3] | Bloomington *Drosophila* Stock Center | RRID:BDSC_5141 | Dr Katja Röper |
| Genetic reagent (*Drosophila melanogaster*) | *patronin*[05252] | PMID:2740359 | | Prof Daniel St Johnston |
| Genetic reagent (*Drosophila melanogaster*) | Patronin-YFP | PMID:2740359 | | Prof Daniel St Johnston |
| Genetic reagent (*Drosophila melanogaster*) | *UAS*::EB1-GFP | PMID:23751496 | | |
| Genetic reagent (*Drosophila melanogaster*) | *Ubi-p63E*::E-cad-GFP | Kyoto Stock Center (DGGR) | RRID:DGGR_109007 | |
| Antibody | Monoclonal mouse anti-α-Tubulin | Sigma-Aldrich | Cat# sc-32293, RRID:AB_628412 | 1:1000 |
| Antibody | Monoclonal rat anti-E-cadherin | Developmental Studies Hybridoma Bank | Cat# DCAD2, RRID:AB_528120 | 1:200 |
| Antibody | Alexa Fluor 488 AffiniPure Donkey Anti-Rat IgG (H+L) | Jackson ImmunoResearch | Cat# 712-545-153 | 1:300 |
| Antibody | Alexa Fluor 594 AffiniPure Donkey Anti-Mouse IgG (H+L) | Jackson ImmunoResearch | Cat# 715-585-151 | 1:300 |
| Antibody | Alexa Fluor 647 AffiniPure Donkey Anti-Mouse IgG (H+L) | Jackson ImmunoResearch | Cat# 715-605-151 | 1:300 |
| Software, algorithm | Tissue Analyser | PMID:20813263 | | |
| Software, algorithm | Microtubule organisation analysis | PMID:27779189 | | |

## Computing the eccentricity of a cell

We uniformly distributed points inside the experimental cell boundary data and used the singular value decomposition on the resulting dataset. The eccentricity then is $ecc = (1 - (a/b)^2)^{1/2}$, where $a<b$ are the singular values.

## Derivation of the *hairyball* distribution

Upon stretching a circular cell by a factor of $b>1$ in the vertical into an ellipse with eccentricity $ecc = (1 - 1/b^2)^{1/2}$, the angles $\theta$ with uniform distribution $R(\theta) = 1/180$ are mapped into the angles $\psi$ with the distribution $\rho(\psi)$ such that $R(\theta) = J\rho(\psi)$, where $J$ is the Jacobian of the stretching map. Since $R(\theta) = const$, $\rho(\psi) \propto J^{-1} = \left(\sin^2(\psi) + b^2 \cos^2(\psi)\right)^{-1}$.

## Parameters of stochastic simulations

We discretized the cell boundary with 180 points, and uniformly placed 200 microtubule minus-ends along the boundary. The short axis of the cell was kept constant at 60 numerical dimer lengths. The critical angle, $\theta_c$, was varied in the range (0, 10, 20, 30, 40) degrees, and the probability of catastrophe, $p_{cat}$, was varied in the range (0.01, 0.1, 0.2, 0.3). The microtubules zipping along cell boundaries were imposed to undergo catastrophe upon reaching the tip of the ellipse or or a sharp cell corner. This mirrors the scenario *in vivo* that due to the high stiffness (*Gittes et al., 1993*), microtubules undergo catastrophe upon reaching an acute cell corner (*Figure 1A*). All the microtubule angle distributions were computed as a time-averaged microtubule dimer angle histograms. For each parameter set, 500 simulations were run to the non-dimensional time $T = 10$, which corresponds to 550 s. At the start of each experiment, all the microtubule lengths were set to zero. During the simulation, the angles with respect to the horizontal of all the dimers of non-zero length microtubules were saved at regular time intervals. We binned these angles into 180 one-degree bins, and then averaged the resulting histograms over the last 2.5 non-dimensional time-units of the simulation, which corresponds to 250 time-points. The *in silico* model is available from the authors upon request.

## Fly stocks

*paired*::Gal4, *engrailed*::Gal4, *UAS*::CD8-Cherry, *UAS*::Khc-RNAi, *Khc*[8], (Bloomington stocks 1947, 30564, 27392, 35770, and 1607, respectively), *UAS*::EB1-DN (**Bulgakova et al., 2013**), *UAS*::Spastin (**Sherwood et al., 2004**), *shot*[3] (gift from K.Röper), *Patronin*[05252], Patronin-YFP (**Nashchekin et al., 2016**), and *Ubi-p63E*::E-cad-GFP (Kyoto DGGR stock 109007). *patronin*, *Khc*, and *shot* mutants are depicted by ⁻ in the text. The flies, embryos, and pupae were kept at 18°C.

## Embryo fixation, pupal wing dissection, and antibody staining

The embryos were fixed as described in **Gomez et al., 2016**. In brief, the staged embryos were dechorionated in 50% bleach for 4 min, and then fixed in 1:1 10% formaldehyde (methanol free, #18814, Polysciences Inc) in PBS:heptane for 20 min at room temperature (RT) and post-fixed/devitellinized for 45 s in 1:1 ice-cold methanol:heptane. Finally, embryos were washed three times in ice-cold methanol, kept in methanol between 6 and 24 hr at −20°C, rehydrated in 1:1 PBS with 0.3% Triton X-100 (T9284, Sigma):methanol, and washed one time in PBS with 0.3% Triton X-100. Rehydrated embryos were blocked for 2 hr in 5% Normal Goat Serum (ab7481, Abcam) in PBS with 0.3% Triton X-100.

Prepupa individuals were collected and aged for 24 hr at 25°C. Pupae were fixed in 10% paraformaldehyde (PFA, #R1026, Agar Scientific) after external cuticle removal for 50 min at room temperature. Then, wings were removed from the carcass, released for their cuticles and post-fixed in fresh 10% PFA for 10 min. Wings were then blocked for 1 hr at room temperature in 10% Native Goat Serum in PBS with 0.3% Triton X-100.

Primary antibody incubations were carried out overnight at 4°C. Primary antibodies used were mouse anti-α-Tubulin 1:1000 (T6199, Sigma), and rat anti-E-cadherin 1:200 (DCAD2, Developmental Studies Hybridoma Bank). In embryos expressing *Khc*-RNAi and all pupal wings, cell boundaries were labelled by native fluorescence of E-cad-GFP instead of antibody. Incubation with secondary antibody was performed for 2 hr at 25°C. Alexa Fluor fluorophore Alexa Fluor 488-, 594- and 647-coupled secondary antibodies (Jackson ImmunoResearch) were used in 1:300. Finally, embryos and wings were mounted in Vectashield (Vector Laboratories).

## Image acquisition

All images were acquired at RT (20-22°C). For quantification of microtubule self-organization, we acquired 16-bit depth images on the Zeiss AiryScan microscope, using a 60x objective lens. For embryos, *z*-stacks of seven sections with 23.5 *px/μm* in XY resolution and 0.38 μm distance between *z*-sections were taken. For pupal wings, z-stacks consisted of five sections with 23.5 *px/μm* in XY resolution and 0.185 μm distance between steps. All processing was done at 6.5 power in the ZEN software. For an analysis of Patronin-YFP distribution, an upright confocal microscope (FV1000; Olympus) using 60 x 1.42 NA oil PlanApoN objective lens was used. 16-bit depth images were taken at a magnification of 12.8 *px/μm* with 0.38 μm between *z*-sections using Olympus Fluoview FV-ASW.

The sub-apical domain of the cell was determined using the E-cadherin junctional signal as a reference for its basal limit, and the absence of α-Tubulin signal as a reference for its apical limit. All imaging was done on dorso-lateral epidermal cells, which excluded the leading edge (first row) cells, given its different identity to the rest of the dorso-lateral epidermal cells. Pupal wings were imaged in their anterior and posterior compartments, the latter detected by expression of mCherry protein. Average projections were made using Fiji (http://www.fiji.sc) for measuring signal area and Patronin-YFP distribution. Figures were assembled using Adobe CS3 Photoshop and Illustrator (http://www.adobe.com). The processing of images shown in figures involved adjusting gamma settings.

## Image analysis to quantify microtubule organization

We used the same workflow as described in the **Gomez et al., 2016**. In short, the E-cadherin signal from a max intensity projection was used to obtain cell outlines using Packing Analyser V 2.0 software (**Aigouy et al., 2010**). These cell outlines were used to identify each cell as an individual object and fit it to an ellipse to calculate eccentricity and the direction of the ellipse major axis using a script in MATLAB R2017b (Mathworks, http://www.mathworks.co.uk). The α-Tubulin signal within each cell was filtered using the cell outline as a mask, and the magnitude of the signal according to its direction analyzed by convolving the filtered α-Tubulin signal with two $5 \times 5$ Sobel operators

(*Gomez et al., 2016*). The resulting magnitude gradient and direction gradient were integrated into a matrix to assign each pixel a direction and magnitude of change in the intensity of α-Tubulin signal. To reduce the noise, the pixels with a magnitude less than 22% of the maximum change were discarded. The remaining pixels were binned with respect to their direction gradient (bin size = 4 degrees). The resulting histogram was normalized. The script is available at https://github.com/nbul/Cytoskeleton; *Płochocka, 2021*; copy archived at swh:1:rev:f94200246cc2389e3c165b5c55bb1ed87ebf8d25.

To calculate the area of α-Tubulin signal within each cell, a custom-made MATLAB script was used. The average projected images were adjusted such that 0.5% of the data with the lowest intensities were set to black and the 0.5% of the data with the highest intensities were set to white in order to compare datasets across genotypes and compensate for potential variability of antibody staining and laser power. The threshold was then calculated using Otsu's method, and a binary image was created using the calculated threshold multiplied by 0.7. This multiplication parameter was determined empirically by testing images from different experiments. Finally, the percent of pixels above the threshold was calculated for each cell. The script is available at https://github.com/nbul/Cytoskeleton.

## Image analysis to quantify Patronin-YFP distribution

The average projection of each *z*-stack was done using Fiji software, and segmented using Packing Analyzer v2.0. The resulting binary images with coordinates of cell-cell borders and vertices were used together with unprocessed average projection by custom-made MATLAB scripts to extract the following values: apical cell area, cell eccentricity, average cell orientation within the image, the orientation of individual cell-cell borders relative to average cell orientation, and mean intensity of each individual border. Only the cells that were completely within the image were taken for quantification. Bristle cells were excluded by their size. Only the borders that are between two cells that were completely within the image were quantified, and the borders adjacent to bristle cells were excluded. The background signal was determined by binarizing images with an adaptive threshold, which uses local first-order image statistics around each pixel and is very efficient in detecting puncta. The mean background signal of cells that were completely within the image was subtracted from the mean intensities of cell-cell borders. The values for each type of border within single embryos were averaged, and the average intensity of borders with 40–90° orientation was divided by the average intensity of borders with 0–10° orientation to produce a single value of asymmetry for each embryo. Finally, to produce distribution of signal intensity by angle, borders of all embryos at stage 15 of development were pulled and binned using 10° bins. The script is available at https://github.com/nbul/Intensity; *Bulgakova and Brown, 2016*.

## Statistical analysis

All data was analyzed using Graphpad Prism 6.0c (http://www.graphpad.com). Samples from independent experiments corresponding to each genotype were pooled and tested for normality with the Shapiro-Wilk test. The α-Tubulin signal area was analyzed using ANOVA with a post-hoc t-test.

## Acknowledgements

We thank the technical staff of the Wolfson Light Microscopy Facility (LMF) and the *Drosophila* Facility at the University of Sheffield for the support with *in vivo* experiments and Prof David Strutt, University of Sheffield, for critical reading of the manuscript. This research was supported by The Maxwell Institute Graduate School in Analysis and its Applications, a Centre for Doctoral Training funded by the UK Engineering and Physical Sciences Research Council grant EP/L016508/01, the Scottish Funding Council, Heriot-Watt University and the University of Edinburgh (AZP); BBSRC BB/P007503/1 (NAB); Royal Society of Edinburgh and the Scottish Government Personal Research Fellowship (LC). Additional support was provided by the Leverhulme Trust grant RPG-2017–249 (to LC and NAB).

## Additional information

### Funding

| Funder | Grant reference number | Author |
|---|---|---|
| Engineering and Physical Sciences Research Council | EP/L016508/01 | Aleksandra Z Płochocka |
| Royal Society of Edinburgh | Royal Society of Edinburgh and the Scottish Government Personal Research Fellowship | Lyubov Chumakova |
| Scottish Government | Royal Society of Edinburgh and the Scottish Government Personal Research Fellowship | Lyubov Chumakova |
| Biotechnology and Biological Sciences Research Council | BB/P007503/1 | Natalia A Bulgakova |
| Leverhulme Trust | RPG-2017-249 | Natalia A Bulgakova Lyubov Chumakova |
| Scottish Funding Council | | Aleksandra Z Plochocka |
| Heriot-Watt University | | Aleksandra Z Plochocka |
| University of Edinburgh | | Aleksandra Z Plochocka |

The funders had no role in study design, data collection and interpretation, or the decision to submit the work for publication.

### Author contributions

Aleksandra Z Płochocka, Software, Formal analysis, Validation, Investigation, Visualization, Writing - original draft; Miguel Ramirez Moreno, Alexander M Davie, Formal analysis, Investigation; Natalia A Bulgakova, Conceptualization, Resources, Data curation, Software, Formal analysis, Supervision, Funding acquisition, Validation, Investigation, Visualization, Methodology, Writing - original draft, Project administration, Writing - review and editing; Lyubov Chumakova, Conceptualization, Resources, Software, Formal analysis, Supervision, Funding acquisition, Validation, Investigation, Visualization, Methodology, Writing - original draft, Project administration, Writing - review and editing

### Author ORCIDs

Aleksandra Z Płochocka (iD) https://orcid.org/0000-0003-4739-1320
Miguel Ramirez Moreno (iD) https://orcid.org/0000-0003-1559-8976
Natalia A Bulgakova (iD) https://orcid.org/0000-0002-3780-8164
Lyubov Chumakova (iD) https://orcid.org/0000-0003-2551-3905

### Decision letter and Author response

Decision letter https://doi.org/10.7554/eLife.59529.sa1
Author response https://doi.org/10.7554/eLife.59529.sa2

## Additional files

### Supplementary files

• Transparent reporting form

### Data availability

At the time of the publication, all the biological data is available on https://doi.org/10.7488/ds/2642.

The following dataset was generated:

| Author(s) | Year | Dataset title | Dataset URL | Database and Identifier |
|-----------|------|---------------|-------------|-------------------------|
| Bulgakova NA, Moreno MR | 2021 | Data file for the submitted publication "Robustness of the microtubule network self-organization in epithelia | https://doi.org/10.7488/ds/2642 | Edinburgh DataShare, 10.7488/ds/2642 |

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

# Appendix 1

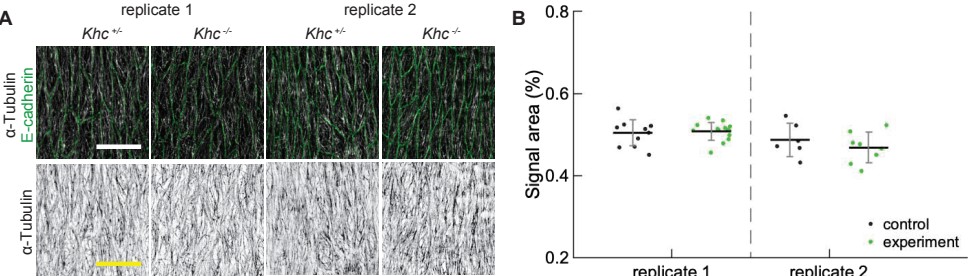

**Appendix 1—figure 1.** Reproducibility of α-Tubulin signal area. (**A**) Apical view of epidermis from heterozygous *Khc* $^{-/+}$ and homozygous *Khc* $^{-/-}$ embryos from two independently repeated experiments (replicates 1 and 2). Cell outlines were visualized by immunostaining against E-cadherin (*green*, top row), and microtubules by immunostaining against α-Tubulin (*white*, top row; *black*, bottom row). Scale bar - 10 μm. (**B**) Quantification of microtubule density in each genotype.

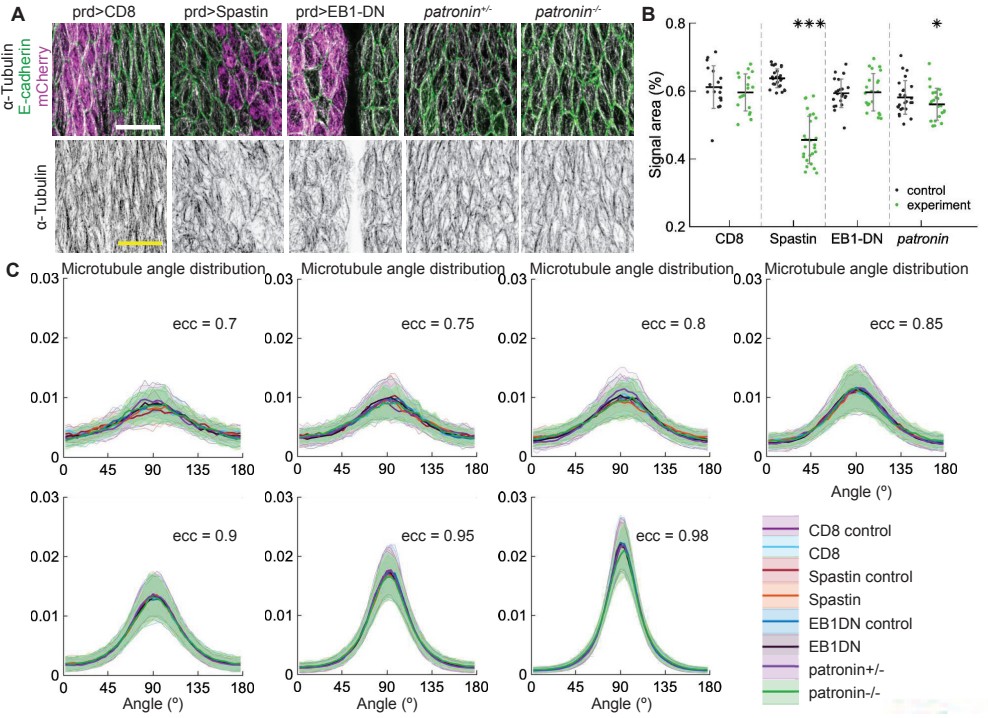

**Appendix 1—figure 2.** Changes to microtubule dynamics and stability do not affect their alignment.
(**A**) Apical view of epidermis from control embryos and with altered microtubules. Left-to-right:
embryos with CD8-Cherry (control), Spastin (Spas), and EB1-DN expressed using paired::Gal4,
heterozygous *Patronin* ⁻/⁺, and homozygous *Patronin* ⁻/⁻ embryos. Cells expressing CD8-Cherry and
EB1-DN are visualized by direct fluorescence of mCherry directly fused to respective proteins,
whereas cells expressing Spastin are visualized by coexpression of CD8-Cherry (*magenta*, top row).
Cell outlines were visualized by immunostaining against E-cadherin (*green*, top row), and
microtubules by immunostaining against α-Tubulin (*white*, top row; *black*, bottom row). Embryos
were imaged across all developmental stages between stages 12 and 15. Scale bar - 10 μm. The
area with no microtubules in the example of EB1-DN expression corresponds to a segmental
groove, where the cells are out of imaging focus. (**B**) Quantification of microtubule density in each
genotype. Internal controls (cells not expressing paired::Gal4) were used for CD8-Cherry, Spastin,
and EB1-DN overexpression. For *Patronin*, heterozygous and homozygous embryos were compared.
*** - $p<0.0001$, * - $p<0.05$ in comparison to respective control. (**C**) The microtubule angle
distributions for each eccentricity (±0.025) do not significantly differ between all genotypes and
relative to controls. The distributions are shown as mean (solid line) with standard deviation
(shading). For each eccentricity, the displayed experimental distribution is the mean distribution
averaged across cells with the set eccentricity (±0.0025 for $ecc = 0.7 - 0.95$ and ±0.005
for $ecc = 0.98$). The number of cells per eccentricity per genotype ranged from 24 to 515.

