## [Decision Letter]

**Acceptance summary:**

Through a combination of experiments, theory, and computation, this paper uncovers several general rules that determine the relationship between cell shape and microtubule orientational order. It exploits the separation of time scales between fast microtubule dynamics (e.g. dynamic instability) and the much slower processes involved in cell shape changes to develop a minimal model that captures a wide range of experimental observations.

**Decision letter after peer review:**

Thank you for sending your article entitled "Robustness of bidirectional microtubule network self-organization" for peer review at *eLife*. Your article is being evaluated by Anna Akhmanova as the Senior Editor, a Reviewing Editor, and three reviewers.

Summary:

Plochocka et al., present a paper that deals with the affect of microtubule polymerization/depolymerization kinetics on the orientation of filaments in large arrays inside a closed cellular geometry. The authors find that the distribution of MT orientations is dependent on cell shape but independent of the kinetic parameters using both experiments and a geometric model of the MT array. Because of this independence, the authors write that this is an archetypal example of robustness in a complex biological system.

Essential revisions:

1) The authors use a purely geometric model of MT growth and the effects on MT kinetics. We are concerned that this is a step too far. First, crosslinking proteins and molecular motors play a huge part in the assemble of MT arrays and these are not considered at all in this work. E.g., does the orientation distribution change if motors are inhibited in the experiments? For the role of motors, see for example:

Zemel, Assaf and Mogilner, (2008).

Zemel, Assaf and Mogilner, (2009).

2) The authors' model treats the MTs as non-interacting which seems incorrect. In addition, it seems that the model is essentially guaranteed to give an alignment along the long axis of an elliptical enclosure because the catastrophe rate is governed by the contact angle between the MT and boundary. The particular form of that rate needs to be explained and justified better. The authors state that the simulation results are insensitive to the alphas and betas, but what about theta_C_? What about proteins that lie on the cell surface and potentially affect MT kinetics in a spatial manner?

3) The authors analyze the MT organization only using the angular distribution across the whole cell. But in the cell, and in the more detailed simulation model, they could look at the local nematic order, and plot its spatial distribution. Surely this will reveal more details, which the detailed simulation should recover better than the "hairyball" model. Similarly, the spatial distribution of the zipped MTs, of the locations of catastrophes etc., as function of eccentricity.

4) The cells are approximated as ellipses, similar to the "spherical cow" simplification. However, cells are usually treated as polygons: do the multi-cell Y-junctions, play a role in the MT organization?

5) The assumption in the "hairyball" model that the MTs remain pointing to the ellipse center is not clear and not explained. is this an effective way to introduce the MT-MT interactions that are missing from this model?

6) In Figure 6 the multiple lines for the cell-shape/ellipse are difficult to discern. We suggest to plot them separately.

Except for the highest eccentricity, the simplest "hairyball" model fits best the experiments. Why? and does it include the non-uniform nucleation of MTs along the membrane?

7) We are concerned that the case for robustness is overstated. The authors should provide justification for the claims that their results (given the caveats above) would be generalizable to other kinds of MT array geometries (as claimed in the discussion). And it is important to emphasize what is being analyzed. The quantities of interest are the AVERAGED distributions, which by definition are taken on time scales long compared to the underlying microscopic dynamics.

As a general principle it is clear that once there is a separation of time scales

then the slow degrees of freedom dominate, and if reorientations occur at the boundary the law of reflection used will dictate the results. That there should be a strong dependence on the cell geometry (i.e. eccentricity) was already implicated in the work of Khuc Trong et al., (where it was MTs emanating from the boundary) and would follow from the dominance of the wall collisions.

In that sense, the present results are reminscent of much older work on wave pattern formation in the Faraday instability Gluckman et al., (1993) in which time averaging of the chaotic patterns revealed mean values that reflected the geometry of confinement. The work also reminds us of that of Dumais et al., on the relationship between cell geometry and cell division planes in plant cells, where microtubule orientations also play a role (Besson and Dumais, (2011)). But it seems to us that a claim of generality must be made after having demonstrated that the issues neglected above do not change the results.

---

## [Author Response]

Essential revisions:1) The authors use a purely geometric model of MT growth and the effects on MT kinetics. We are concerned that this is a step too far. First, crosslinking proteins and molecular motors play a huge part in the assemble of MT arrays and these are not considered at all in this work. E.g., does the orientation distribution change if motors are inhibited in the experiments? For the role of motors, see for example:Zemel, Assaf and Mogilner, (2008).Zemel, Assaf and Mogilner, (2009).

We thank the reviewers for raising this important point. Indeed, we are aware of the role of molecular motors and crosslinkers in MT organization. While the model we use in stochastic simulations is purely geometrical, it includes the effect of crosslinking proteins via a large-scale parameterization of MT-MT interactions. It is done by introducing the probability of a polymerizing MT zipping along the MT it encounters, which depends on the angle between these MTs. Here, the concentration and the nature of the crosslinking proteins affect theta_c_ and p_cat_. We show in Figure 2 that varying these two parameters does not affect the average MT angle distribution, although it affects the distribution of MT density per unit area inside individual cells (see the new Figure 3). The case of theta_c_=0 corresponds to non-interacting MTs and could be interpreted as a model of an experiment in the absence of crosslinkers.

We expect that in the studies mentioned in the question, the effect of the crosslinkers is much stronger than in our system. In these papers, due to the fact that the MTs are free (as opposed to anchored) and stabilized (as opposed to dynamic) with either free or periodic boundary conditions, the crosslinkers play a dominant role in the MT organization. However, in our system, the MT minus-ends are anchored on the cell boundary, while the MT plus-ends are dynamic. Anchoring inhibits MTs sliding along each other in bundles, and dynamic plus-ends allow MTs to “sense” the scale of the cell by growing towards another cell boundary and, upon reaching it, either growing parallel to it or collapsing. Due to these two effects, the role of crosslinkers in our system is secondary, which we now support by *in vivo* experiments. For this reason, we did not directly include sliding as an additional degree of freedom in our mathematical model. We now mention the roles of crosslinkers and molecular motors in the Introduction; the clarifications of how actions of motors and crosslinkers are indirectly incorporated into the model; and the aforementioned interpretation of the results with theta_c_=0 in subsection “Stochastic simulations demonstrate robustness of microtubule self-organization for a wide range of parameter values”.

Supporting robustness of MT self-organization and in agreement with our model, we show now that *in vivo* the loss of zygotic kinesin-1 and its knockdown with RNAi -- the motor protein known to “bundle” MTs (e.g., Yan et al., 2013) -- did not affect MT angle distribution in our experimental system. Although testing all possible candidates is outside of the scope of this paper, we also demonstrated that the loss of zygotic MT-actin crosslinker Shortstop -- a protein known to regulate MTs in the *Drosophila* epidermis (Takacs et al., 2017) -- similarly did not affect MT angle distribution, further substantiating our findings. These new results are included in subsection “*in vivo* manipulations of microtubule dynamics and stability alter microtubule density but not alignment”, in the new Figure 4, and in the Discussion section.

(2.1) The authors' model treats the MTs as non-interacting which seems incorrect.

We would like to highlight that the detailed stochastic simulations were performed with the interacting MTs, where the parameterization of the MT-MT interaction was inspired by that used in plants (Tindemans et al., 2010; Allard et al., 2010). Furthermore, the simplified analytical model with non-interacting MTs was only proposed after we discovered that the MT angle distribution is independent of the MT-MT interactions in the detailed simulations and confirmed this *in vivo*.

(2.2) In addition, it seems that the model is essentially guaranteed to give an alignment along the long axis of an elliptical enclosure because the catastrophe rate is governed by the contact angle between the MT and boundary.

We agree with the reviewer that the angle-dependence of the MT catastrophe rate at the boundary ensures alignment with the long axis of an elliptical enclosure. However, it does not explain how well individual MTs are aligned with each other -- the full MT angle distribution -- or how it depends on the cell geometry in non-elliptical enclosures. We have previously shown that cell geometry guides MT selforganization in epithelia *in vivo* (Gomez et al., 2016), and hence made understanding the nature of the full MT angle distribution the focus of this paper. We highlighted this point subsection “An analytical model shows that microtubule self-organization depends on the cell geometry and minus-end distribution”. We found the analytical form of the MT angle distribution, how it depends to leading order on cell geometry and weakly depends on the MT dynamic instability rates. This analytical result further predicted that the density of the MT minus-ends and the average cell-shape are required to obtain a quantitative agreement with the *in vivo* experimental results. We would add here that the alignment with the cell major axis is indeed guaranteed, and it contributes to the robustness phenomenon. This is one of the main points of our work, highlighting the independence of the average MT organization of most of the details, such as the dynamics of individual MTs.

Finally, we would like to mention that after receiving the reviews, we appreciate that the title could be misleading. We renamed the paper so that the title reflects the findings more precisely as "Robustness of the microtubule network self-organization in epithelia”.

(2.3) The particular form of that rate needs to be explained and justified better.

This form of the rate was originally inspired by the well-established induction of catastrophe when a MT grows against a barrier (Janson et al., 2003). The role of cell borders as barriers in epithelial cells is supported by an observation that MTs buckle at the cell cortex (Singh et al., 2018). Finally, in our previous work, we discovered experimentally that the MT catastrophe rate at the cell boundary *in vivo* is angle-dependent (see Figure 5 in Gomez et al., 2016). Altogether, we are confident that the use of this particular form of MT-border interaction is well-justified and closely reflect this interaction *in vivo*. We thank the reviewers for this comment and have included additional justification in subsection “Stochastic simulations demonstrate robustness of microtubule self-organization for a wide range of parameter values”.

(2.4) The authors state that the simulation results are insensitive to the alphas and betas, but what about theta_C_?

Since MTs are stiff filaments, we only tested the cases of theta_c_ between 0 to 40 degrees (at 0, 10, 20, 30, and 40 degrees) to remain within the biologically relevant regime (Dixit and Cyr, 2004). We reported our results in Figure 2, where we chose to only plot them for the smallest non-zero and the largest values of theta_c_ (10 and 40 degrees) because there was no significant difference in the MT angle distribution curves. The case of theta_c_=0 is the case of non-interacting MTs, for which the MT angle distribution is given by the analytical model presented later in the paper. We clarified this in subsection “Stochastic simulations demonstrate robustness of microtubule self-organization for a wide range of parameter values”. We also included a new Figure 3, where we show how local microtubule density depends on theta_c_, in contrast to the statistics captured by the MT angle distribution, which is insensitive to the strength of MT-MT interaction.

(2.5) What about proteins that lie on the cell surface and potentially affect MT kinetics in a spatial manner?

Here we interpret “the cell surface*”* in the question to refer to the apical membrane. Although we do not know which particular proteins the reviewers refer to, we tested effects of shotstop loss and did not see any changes of average MT alignment although the local organization appears altered – as expected, MTs are more likely to run along cell boundaries. We included a description of this finding in subsection “*in vivo* manipulations of microtubule dynamics and stability alter microtubule density but not alignment”. The discovery of other proteins “that lie on the cell surface and potentially affect MT kinetics in a spatial manner” is a subject of future studies and is outside of the scope of this paper.

At the same time, we have recently discovered that molecules localized in the plane of MTs (e.g.actin cables) alter the MT self-organization (Płochocka et al., 2019). The main effect of actin cables was to shift the mean of the MT angle distribution towards their direction. We included a more extensive discussion about the effects of spatially distributed regulators and crosslinkers on the MT self-organization and robustness in subsection “The importance of microtubule interactions”.

3) The authors analyze the MT organization only using the angular distribution across the whole cell. But in the cell, and in the more detailed simulation model, they could look at the local nematic order, and plot its spatial distribution. Surely this will reveal more details, which the detailed simulation should recover better than the "hairyball" model. Similarly, the spatial distribution of the zipped MTs, of the locations of catastrophes etc., as function of eccentricity.

We appreciate the reviewer’s question. Our main reason for focussing on the averaged qualifiers of the MT network was to enable comparison of our computational and analytical results with *in vivo* data. Since it is noisy, we could only perform comparisons of the averaged quantities. At first, we attempted to use the nematic order parameter S_2_ to quantify the degree of alignment in the detailed simulations, but the density of MTs in the simulations was too low to produce promising results. This was due to the fact that in our system the rescue rate at the MT minus-ends on the cell boundary is low, since it mimics the situation *in vivo*, where the rescue rate is the same on the boundary as in the cell interior. We, therefore, returned to working with the averaged MT angle distribution to quantify the MT alignment. We have included this clarification the Introduction.

As per reviewer’s request, we have included additional results of the stochastic simulations. In particular, we have included how local MT density is affected by MT zipping in a new Figure 3. We showed how MT density changes for a cell of a given eccentricity with increasing theta_c_ for a fixed value of p_cat_, since we found that the degree of MT zipping increases with increasing theta_c_ and is not significantly altered by changing p_cat_ in the range used in our simulations. We have added the description of these results in subsection “Stochastic simulations demonstrate robustness of microtubule self-organization for a wide range of parameter values”. Our experimental results further show that *in vivo* zipping varies (Figure 4), and that loss of the crosslinker shot seems to lead to more zipping near cell boundaries consistently with a reduced theta_c_. We have added a discussion of these findings in subsection “Robustness on the tissue scale”, and in subsection “The importance of microtubule interactions”.

Finally, we did not include the spatial map of averaged zipping in the detailed simulations since it was not very insightful - zipping increases with theta_c_, and on average more zipping happens closer to the cell center. We also did not include the map of catastrophes since they mostly occur at the cell boundary.

4) The cells are approximated as ellipses, similar to the "spherical cow" simplification. However, cells are usually treated as polygons: do the multi-cell Y-junctions, play a role in the MT organization?

We agree with the reviewers that the cells are polygons. As the analytical MT angle distribution in Equation 11 shows, the time-averaged MT angle distributions are robust but differ between cells, as they depend on the geometry of the cell boundary. When we computed the MT angle distributions in cells with corners, the MT angle distribution had peaks corresponding to the corners. This is due to the fact that to leading order, the MT angle distribution is proportional to the square of the cell crossection integrated over the cell boundary. However, *in vivo*, this effect averages out for the tissue-averaged MT angle distribution. We hypothesize that the Y-junction geometry matters for local processes that rely on the MT dynamics, particularly on the MT plus ends, and indeed we observed local irregularities in their organization. However, this was not the focus of this manuscript. We elaborated on this point in subsection “The analytical model accurately predicts microtubule-self organization given both the experimental cell shape and distribution of microtubule minus-ends”, and in subsection “The importance of microtubule interactions”.

5) The assumption in the "hairyball" model that the MTs remain pointing to the ellipse center is not clear and not explained. is this an effective way to introduce the MT-MT interactions that are missing from this model?

We thank the reviewers for the comment. We would like to emphasize that the *hairyball* model is a 0-th order conceptual model that cannot include MT-MT interactions. The primary purpose of considering this model was to address the question we were often asked: “Would stretching the cell give you the experimental result?”. The answer was surprising -- it would for eccentricity up to 0.95. However, this model's assumptions are not physical: the MTs are not dynamic, and their average directions pointing to the center of the cell after the cell stretching is a byproduct of stretching both the cell and the MT angle distribution in the same manner. We now clarify this in subsection “A conceptual geometric model accurately predicts *in vivo* alignment”.

It is an interesting suggestion that perhaps “MTs remain pointing to the ellipse center” is a “way to introduce the MT-MT interactions”. However, in the stochastic simulations, the MT angle distribution does not depend on the parameterization of MT-MT interactions; therefore, it is probably not why the distribution is a good fit for the experimental data.

The main benefit of including this conceptual model in the paper is that it provides the cell biology community with a simple formula to fit the MT angle distribution in epithelial cells with only one-parameter - the “effective aspect ratio”. This fit works much better than other widely used distributions such as von Mises distribution. Additionally, the fact the hairyball distribution does not depend on the MT dynamic instability parameters supports the robustness conclusion, further highlighting that geometry is one of the most significant determining factors of the MT self-organization.

Reading the reviews, we realize that perhaps it would improve the paper if we emphasized that the model is conceptual in the subsection title. Hence, we changed the corresponding subsection's title to “A conceptual geometric model accurately predicts *in vivo* alignment” and highlighted the model’s usefulness to the community of biologists both in subsection “A conceptual geometric model accurately predicts *in vivo* alignment”and in the Discussion.

(6.1) In Figure 6 the multiple lines for the cell-shape/ellipse are difficult to discern. We suggest to plot them separately.

As we state in the manuscript, it is not known why the hairyball model fits the experiments up to ecc=0.95. We tried to find if the resulting distribution asymptotically matches the analytical formula or has underlying physical significance, and we did not find such connections.

As stated in the manuscript, the results of the hairyball model do not include non-uniform nucleation of MTs along the membrane. However, the non-uniform minus-end density could be easily included as a multiplicative factor. We tested it, and it did not change the agreement with the experimental data. We noted this in the Discussion. That said, we would prefer not to include in detail the case of non-uniform nucleation since this geometric model is already purely conceptual and excludes the MT dynamics.

We would also like to add that we tried other conceptual models but did not succeed. In particular, we tested if the length-averaged angle distribution of random “sticks” placed inside an ellipse explains the *in vivo* MT angle distribution. This tests the hypothesis that it is possible to have the experimental MT angle distribution in the absence of internal MT alignment and only due to elongated cell geometry. This did not work, and hence there is something about the stretching deformation used in the hairyball distribution that captures the fact that the MTs “sense” the cell geometry due to the fact that they are dynamic and anchored at the cell boundary. We decided not to include other conceptual models in the manuscript as they did not produce meaningful results.

(6.2) Except for the highest eccentricity, the simplest "hairyball" model fits best the experiments. Why? and does it include the non-uniform nucleation of MTs along the membrane?

We altered this figure as suggested, which is now the new Figure 8.

7) We are concerned that the case for robustness is overstated. The authors should provide justification for the claims that their results (given the caveats above) would be generalizable to other kinds of MT array geometries (as claimed in the discussion). And it is important to emphasize what is being analyzed. The quantities of interest are the AVERAGED distributions, which by definition are taken on time scales long compared to the underlying microscopic dynamics.As a general principle it is clear that once there is a separation of time scalesthen the slow degrees of freedom dominate, and if reorientations occur at the boundary the law of reflection used will dictate the results. That there should be a strong dependence on the cell geometry (i.e. eccentricity) was already implicated in the work of Khuc Trong et al., (where it was MTs emanating from the boundary) and would follow from the dominance of the wall collisions.In that sense, the present results are reminscent of much older work on wave pattern formation in the Faraday instability Gluckman et al., (1993) in which time averaging of the chaotic patterns revealed mean values that reflected the geometry of confinement. The work also reminds us of that of Dumais et al., on the relationship between cell geometry and cell division planes in plant cells, where microtubule orientations also play a role (Besson and Dumais, (2011) ). But it seems to us that a claim of generality must be made after having demonstrated that the issues neglected above do not change the results.

We agree with the reviewer that “the quantities of interest are the AVERAGED distributions” and reemphasized it in Introduction; in the text in subsection “Stochastic simulations demonstrate robustness of microtubule self-organization for a wide range of parameter value”, subsection “*in vivo* manipulations of microtubule dynamics and stability alter microtubule density but not alignment”, subsection “The analytical model accurately predicts microtubule-self organization given both the experimental cell shape and distribution of microtubule minus-ends”; and in Discussion.

We also appreciate the discussion of similarities between the results in different fields where the average of fast small-scale dynamics strongly depends on the slow degrees of freedom, which is the cell geometry in epithelia. We expect that since the systems discussed in subsection “Our system is a particular but generalizable scenario” are similar to the one we have considered, the robustness result holds, because in all of them the MTs are dynamic and anchored, while the separation of time-scales is also valid. We clarified this further in the manuscript in subsection “Our system is a particular but generalizable scenario”.

Since it is impossible to test the results in all the possible MT array geometries, the authors meant the mentioned conclusion of the paper to be a suggestion (not a claim). However, we now support the findings of this paper with experiments in an additional tissue – pupal wings – subsection “*in vivo* manipulations of microtubule dynamics and stability alter microtubule density but not alignment”, and new Figure 5. Therefore, while we tuned down this statement, we are confident about the generality of our result.